

# Scale patterns of the Sentinel-1 SAR-based snow depth product compared to station measurements and airborne LiDAR observations

Jiajie Ying[1], Lingmei Jiang[1], Jinmei Pan[2], Chuan Xiong[3], Jianwei Yang[1]

[1]State Key Laboratory of Remote Sensing Science, Jointly Sponsored by Beijing Normal University and Aerospace Information Research Institute of Chinese Academy of Sciences, Faculty of Geographical Science, Beijing Normal University, Beijing 100875, China
[2]China National Space Science Center, Chinese Academy of Sciences, Beijing 100190, China
[3]Faculty of Geosciences and Environmental Engineering, Southwest Jiaotong University, Chengdu 610031, China

*Correspondence to*: Jianwei Yang (yangjw@bnu.edu.cn)

**Abstract.** Water storage in snowpacks in mountain areas is critical for hydropower production, hydrological forecasting, and freshwater availability. Spaceborne synthetic aperture radar (SAR) is a powerful tool for quantitatively measuring snow mass because of its high spatial resolution and the sensitivity of signals to snow depth (SD). In particular, the first SAR SD product (C-snow) based on Sentinel-1 satellites displays high sensitivity to depolarization signals for dynamic SD monitoring in mountainous areas. Moreover, upscaled C-snow retrievals (e.g., 10 and 25 km) have been used to provide reference data to train machine learning models, improve passive microwave-based retrieval, and calibrate many hydrological models. However, a systematic assessment of C-snow products at various scales has not been conducted, until now. In this study, the performance of C-snow products at three scales (1, 10 and 25 km) is comparatively assessed via station-based measurements and airborne LiDAR observations, and the scale patterns associated with the heterogeneity of the geographic environment and the representativeness of so-called truth data are analyzed. The results indicate that the scale patterns of the C-snow products across various resolutions differ from those of station- and airborne-based reference data. As the spatial scale increases from 1 to 25 km, the error of C-snow retrieval in reference to station measurements tends to increase (e.g., ubRMSE from 68.18 to 77.47 cm, bias from -9.81 to 10.68 cm), whereas it tends to decrease compared with airborne snow observatory (ASO) data, with ubRMSE values ranging from 104.3 to 83.29 cm, and the bias values from -91.31 to -52.73 cm. We also found that land cover types, e.g., tree cover and permanent ice, affect the C-snow product at various scales. Especially an overestimation tends to occur in coarse pixels covered with even a small amount of permanent ice. It is concluded that C-snow retrieval at three scales is characterized by high uncertainty. Researchers should focus on developing a robust SD retrieval algorithm by combining SAR backscattering signals and polarimetric and interferometric information.





## 1 Introduction


Snow storage and seasonal meltwater in mountains are components of the "water towers" that form in global mountains. Therefore, quantitatively estimating snow mass in mountain areas is very important for hydropower production, hydrological forecasting, and freshwater availability (Barnett et al., 2005; Dozier et al., 2016; Daloz et al., 2020; Qin et al., 2020). The snow water equivalent (SWE) is a parameter that reflects how much water the snowpack contains, which typically can be

estimated from snow depth (SD). Satellite remote sensing has been demonstrated to be an effective tool for monitoring multiscale SD information, which enhances our understanding water availability in snowpacks (Chang et al., 1987; Kelly, 2009; Takala et al., 2011; Lievens et al., 2019).

Microwave remote sensing is the most widely-used technology for retrieving SWE because of its penetrating ability to the snowpack and the volume scattering effects caused by snow particles (Chang et al., 1987; Tsang et al., 2022). Generally,

active microwave remote sensing, especially synthetic aperture radar (SAR), has advantages over passive microwave techniques (e.g., radiometer-based methods) for characterizing the SWE across high-mountain regions (Dozier et al., 2016). Notably, spaceborne SAR can support fine-spatial-resolution (dozens of meters) monitoring compared with passive microwave remote sensing (dozens of kilometers). Additionally, the snow cover in mountain areas is typically deep (up to meters), and snowpack evolution in these areas is generally much more complex than that in flat areas. For example, the

snow density can reach 550–700 kg/m3 due to snowfall accumulation and wind- and gravity-driven compaction (Lemmetyinen et al., 2016; Venäläinen et al., 2021). In addition, owing to the large negative temperature gradient between the air temperature and ground temperature, the development of depth hoars is common (Fierz et al., 2009). For example, the snow grain size in depth hoars can reach the centimeter level at high elevations and on shady slopes (King et al., 2018; Picard et al., 2022). Thus, the signals of typically used frequencies (e.g., the Ka-band) in passive microwave remote sensing

tend to be saturated within the SD range of 40–80 cm (Derksen et al, 2010; Takala et al., 2011; Picard et al., 2018).

In recent years, the scientific community has focused on the monitoring SWE in mountain regions via C-band SAR observations, benefitting from their strong penetration depth and data accessibility. Notably, although snow volume scattering is stronger in high-frequency Ku-bands than in other bands in theory, the sensitivity of the backscattering coefficient at this frequency is also limited to approximately 150 cm (Rott et al., 2010; Cui et al., 2016; Zhu et al., 2021).

Moreover, Lievens et al. (2019) observed the sensitivity of the depolarization signal (VH/VV) in the C-band to the SD and innovatively developed a C-band-based SD retrieval algorithm for mountains in the Northern Hemisphere. Notably, the backscattering coefficient at cross-polarization is more sensitive to volume scattering than co-polarization is due to the non-spherical properties of snowpack, and this physical mechanism is used for SD retrieval; in addition, co- and cross-polarization signals are similar for surface scattering at the snow–soil boundary (Shi and Dozier, 2000; Lievens et al., 2022;

Borah et al., 2024). Thus, the ratio of cross- to co-polarization signals enhances the snow volume scattering and weakens the surface scattering between snow and soil.



Given its high resolution of 1 km, the C-snow product can potentially be used to assess the heterogeneity of the snow distribution in mountain areas (Alfieri et al., 2022; Girotto et al., 2024). However, it has only been evaluated from the point to regional scales until now, not at the global scale. For example, an evaluation across the Po River basin in Italy revealed that the RMSE ranges from 20 to 60 cm, in reference to ultrasonic sensor measurements (Alfieri et al., 2022). Sourp et al. (2024) compared C-snow retrieval products with the results of airborne lidar surveys in the Sierra Nevada region from 2017–2019 and reported that the RMSE ranged from 21 to 138 cm, and that the bias reached up to -124 cm. Hoppinen et al. (2024) also evaluated algorithm performance at six study sites across the western United States on the basis of 2020-2021 airborne LiDAR observations, with mean RMSE and bias values of 92 cm and -49 cm, respectively. In addition, C-snow products at 10 and 25 km resolutions have been used as reference datasets, such as for training samples for machine learning models to improve passive microwave SWE estimates (Xiong et al., 2022; Broxton et al., 2024; Yang et al., 2024). Lievens et al. (2022) utilized Sentinel-1 observations to monitor SD in the European Alps and evaluated its performance across different resolutions. When compared to the 500 m and 1 km resolutions (by linearly averaging the 100 m retrievals), the performance at 100 m resolution showed slight degradation, due to the impacts of radar speckle noise, geometric distortions, and local heterogeneity in topography, land surface properties, and snow characteristics. The accuracy of the C-snow product at 1-, 10- and 25-km still requires further investigation. The scale effect across three spatial resolutions and sensitive influencing factors (e.g., topography, land cover and wet snow) are crucial to consider when evaluating the performance of C-snow product. However, exploration of these factors remains insufficient, thus hindering our understanding of ways to further improve C-band SD retrieval technology.

Therefore, the specific objectives of our study are to (1) systematically evaluate the error of C-snow retrieval across 1-, 10- and 25-km spatial scales via both ground-based measurements and airborne LiDAR data and analyze the sensitivity of the error to various factors, as well as (2) quantitatively compare the scale patterns of the SD products at three spatial scales and explain the inconsistency of scale effects on the basis of different reference datasets (stations vs. ASO). To achieve this goal, we used measurements from point-scale stations and airborne LiDAR campaigns. The latter has a much wider global coverage, whereas the former is ideal for characterizing the SD distribution and assessing snow heterogeneity. This paper is structured with five sections. Section 2 describes the methods and data. The results and discussion are presented in Section 3 and Section 4, respectively. Finally, Section 5 presents the conclusions, and future research is discussed.

## 2 Data and Methodology

### 2.1 Sentinel-1 SD product

The first 1-km SD product based on C-band SAR covers all mountain ranges in the Northern Hemisphere and can be downloaded from https://ees.kuleuven.be/project/c-snow. An empirical change detection method is used to retrieve SD (Lievens et al., 2019). The available C-snow dataset covers the period from September 1, 2016, to May 19, 2019. The spatial





resolution of the C-snow product is 1 km, and the temporal resolution varies from daily to every two weeks, depending on the frequency of Sentinel-1 observations.

## 2.2 Reference SD data

### 2.2.1 Ground-based measurements

In this study, we collected six ground-based observational datasets as reference data to evaluate and compare the performance of the C-snow product across three scales (Figure 1). They include the Global Historical Climate Network (GHCN), Canadian Historical Snow Water Equivalent (CanSWE), in situ measurements from Chinese weather stations (China-SD), SD measurements from Maine (Maine-SD), SWE variables from Snow Telemetry (SNOTEL) and an SWE dataset in the range of the former Soviet Union (Russia-SWE). For the SNOTEL and Russia-SWE datasets, we used a fixed snow density of 0.24 g/cm³ to convert the SWE to SD (Takala et al., 2011; Luojus et al., 2021).

The ground-based observations span multiple regions and stations. The CanSWE dataset in Canada includes data from 273 stations in mountain regions and can be accessed via https://zenodo.org/records/5217044#.YdYEsllybb0 (Vionnet et al. 2021). The GHCN dataset includes data from 4,133 stations in mountain regions and provides SD values worldwide (ftp://ftp.ncdc.noaa.gov/pub/data/ghcn/daily/) (Menne et al. 2012). The China-SD dataset from the China Meteorology Administration (http://data.cma.cn/) includes observations from 744 stations in mountainous regions. The SNOTEL dataset was acquired from 677 stations in mountainous regions in the United States (https://toolkit.climate.gov/tool/snow-telemetry-snotel-data-viewer). The Russia-SWE dataset from former Soviet Union regions contains observations from 52 stations in mountain regions, and it can be downloaded from the All-Russia Research Institute of Hydrometeorological Information–World Data Center (http://meteo.ru/english/climate/snow1.php). Additionally, the Maine-SD dataset for the Maine region includes information from 92 stations in mountainous regions; it can be accessed via https://mgs-maine.opendata.arcgis.com/datasets/maine/about. Figure 1 shows the distribution of stations for these datasets.

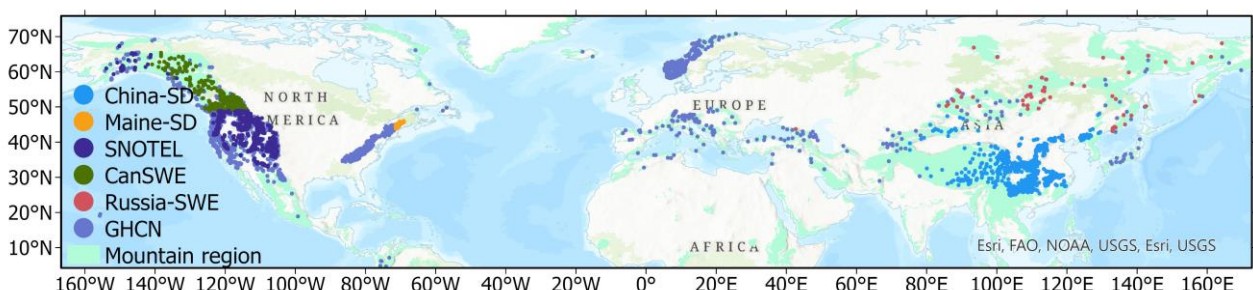

**Figure 1.** Spatial distribution of stations in various SD datasets.

To control the quality of the station observations, we excluded observations with SD of zero, removed observations higher than twice the 95th percentile, and excluded stations with fewer than three SD measurements over the entire study period. Later, we calculated the averages of measurements from multiple stations within pixels at three scales of 1, 10 and 25

km (see details in Section 2.4). Figure 2a-Figure 2c show the processed grids of SD as the reference datasets at the 1-, 10-
120 and 25-km scales in two selected areas; the number of grids decreases from 2881 to 2231 and to 1503 grids as the scale
increases.

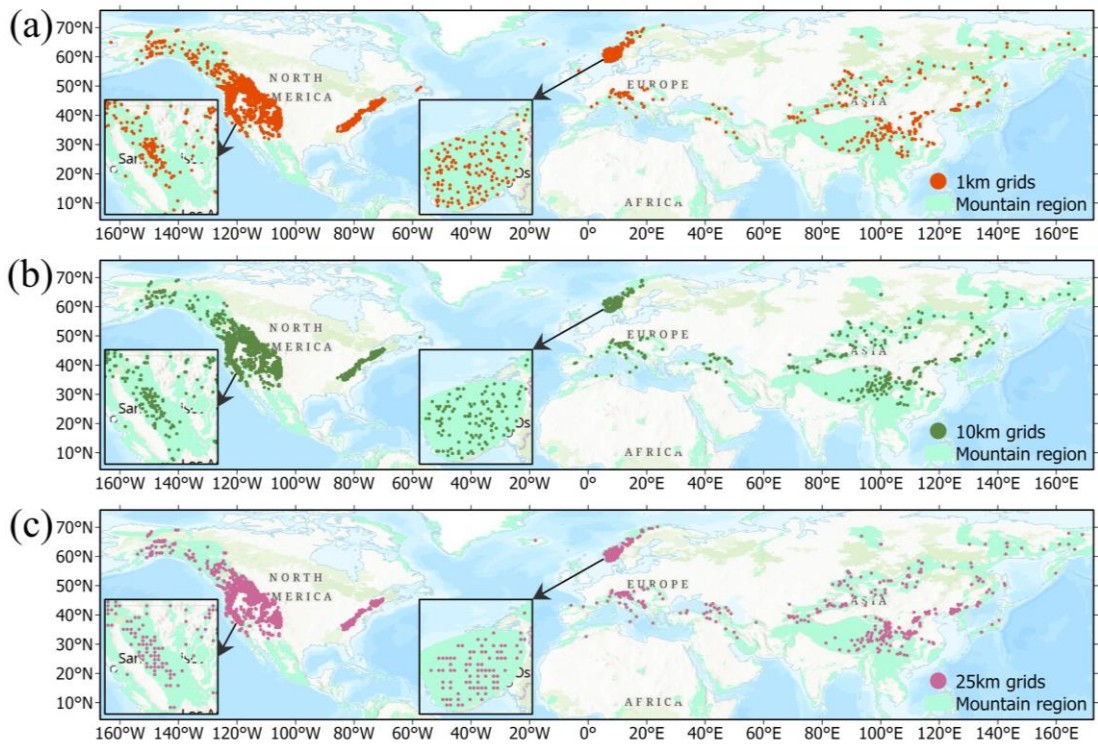

**Figure 2.** Spatial distributions of the matched grids at (a) 1-, (b) 10-, and (c) 25 km scales. Zoomed-in views show the detailed
distributions of grid locations in the Sierra Nevada range over the United States and the Jotunheimen mountain range in Norway and
125 Sweden.

### 2.2.2 Airborne snow observatory (ASO) data

The ASO is a LiDAR mission designed to measure the evolution process of SD during different snow seasons in major
watersheds in the western United States (Painter et al., 2016). Airborne remote sensing campaigns over the western United
States were conducted from 2013 to present, and hyperspectral reflectance and LiDAR SD data were collected in Colorado,
California, Oregon, and Washington. These data are used to develop standard basin-scale instantaneous SD maps, with a
resolution of 3 m and an evaluated accuracy of 0.08 m (Painter et al., 2016). To assess and compare the accuracy of the C-
snow product at different scales, we obtained 59 ASO maps at a 3 m resolution from September 2016 to May 2019 (Figure
3). Figure 4 provides the statistics of the available measurements by basin and date.



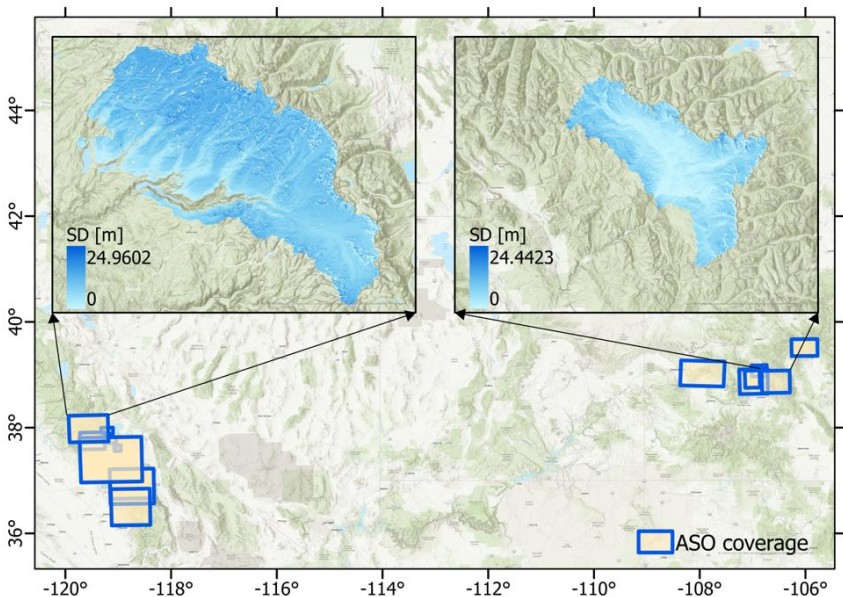

**Figure 3.** Spatial distributions of the ASO coverages in this study. The left subplot presents the SD distribution in the Tuolumne Basin on March 24, 2019, and the right subplot displays the SD distribution in the Gunnison-Taylor River Basin on March 30, 2018.

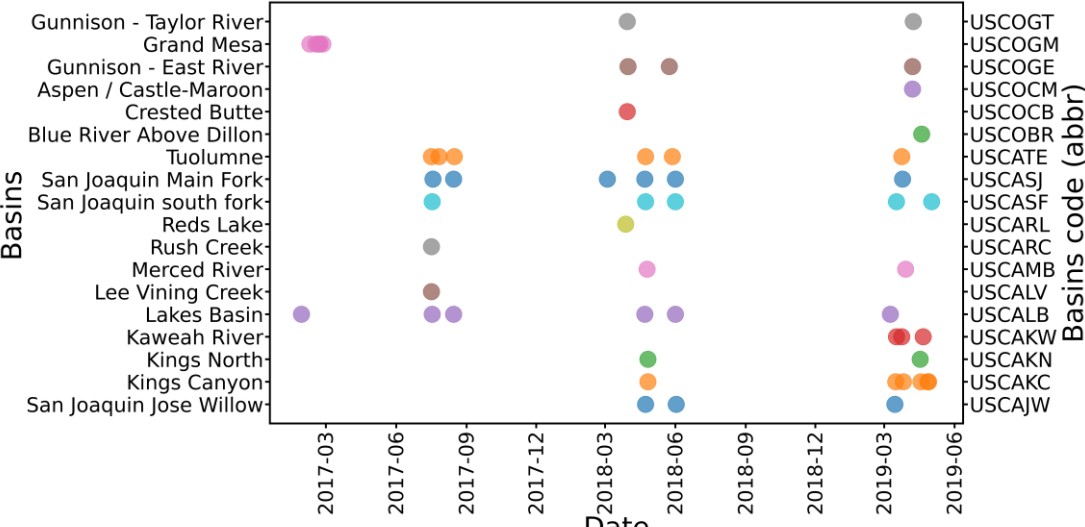

**Figure 4.** Temporal distribution of the ASO observations used in this study.

## 2.3 Auxiliary data

To evaluate the influence of land cover type, forest fraction, and topography (elevation and its standard deviation) on the accuracy of C-snow SD, we collected corresponding datasets from the Google Earth Engine (https://earthengine.google.com) and processed them at various scales (1, 10, and 25 km). Table 1 lists the auxiliary data used in this study.



To match the scales of the C-snow SD data, we resampled the auxiliary data accordingly. The land cover type data were
first resampled to a 1 km resolution via the mode resampling method. Subsequently, these 1 km land cover data were further
resampled to 10- and 25-km resolutions. During the resampling process, if a certain land cover type occupied 80% or more
of a large grid of 10 or 25 km, that type was then assigned to the resampled grid; otherwise, no land cover type was assigned.
The forest fraction, elevation, and standard deviation of elevation at a 1 km resolution were resampled to 10 and 25 km
resolutions via the average resampling method.

**Table 1.** Description of the auxiliary data used in this study.

| Name | Source | Initial resolution | Coarse resolution |
| --- | --- | --- | --- |
| Land cover type | European Space Agency (ESA) WorldCover 10 m 2020 product (Zanaga et al., 2021) | 10 m | |
| Forest fraction | | 10 m | |
| | | | 1, 10, and 25 km |
| Elevation | Multi-Error-Removed Improved-Terrain (MERIT) DEM (Yamazaki et al., 2017) | 3 arc second | |
| Standard deviation of elevation | | | |

## 2.4 Methodology

Figure 5 shows the workflow of this study. To assess the accuracy of C-snow retrieval at different scales, the 1 km C-
snow product was resampled to 10 and 25 km, respectively. Here we directly used the mean resampling method according to
previous studies (Broxton et al. 2024; Herbert et al. 2024). Meanwhile, we tested and compared the mean and median
sampling methods, and found the validation results were similar. To control equality and the representativeness of coarse-
resolution pixels, we selected only the 10- and 25-km grids in which the percentage of the snow-covered area was at least
80%. The average 1 km SD values within the coarse-resolution pixels were then used as the 10- or 25-km-scale products
(Figure 5). To compare C-snow with ground-based data and ASO observations at different scales, we also resampled the 3-
m-resolution ASO data to 1, 10, and 25 km. When resampling the ASO data, we only calculated the average for grids for
which the number of 3 m ASO observations at the larger scales (1, 10, and 25 km) was not less than 30%. Here, we selected
30% because it ensures a sufficient quantity and representativeness of validation samples.

Additionally, we explored the relations between land cover type, forest fraction, elevation and standard deviation of
elevation on C-snow SD across different scales. Four evaluation metrics were used to assess the C-snow products: the
correlation coefficient (corr.coe), bias, unbiased root mean square error (ubRMSE), and relative bias (Rbias).





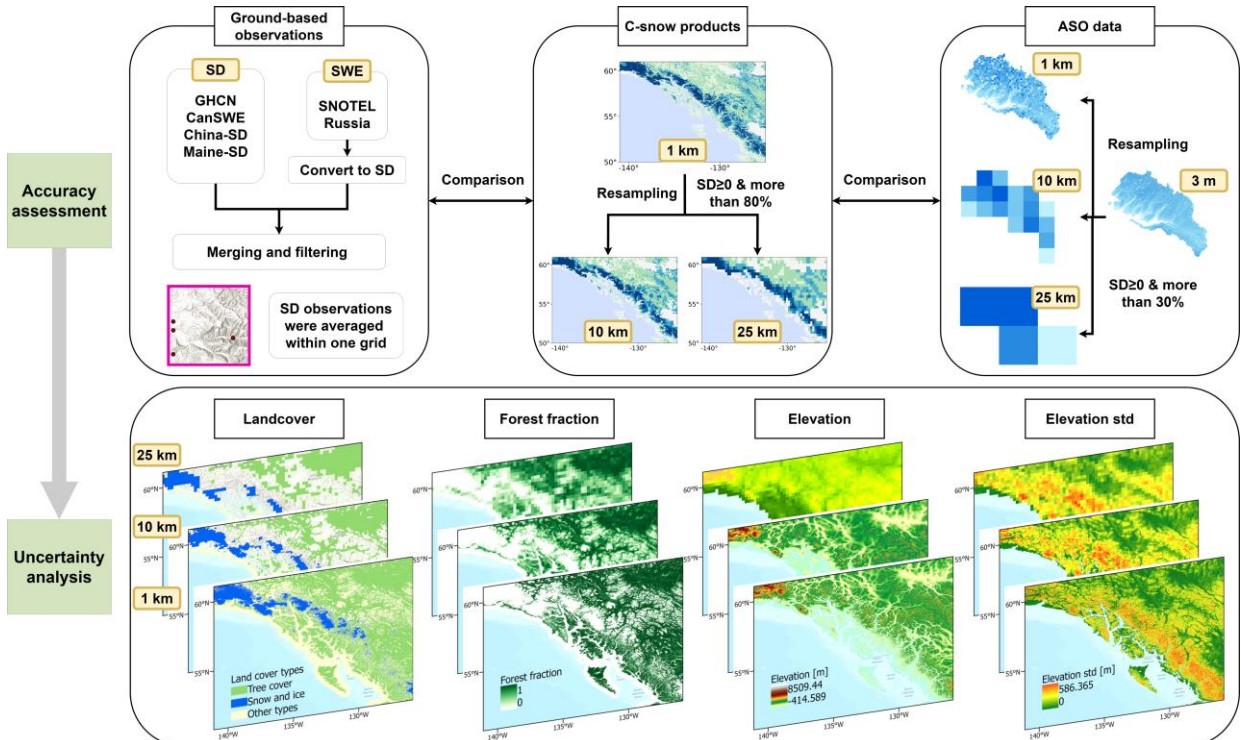

**Figure 5.** Diagram of the research workflow.

## 3 Results

### 3.1 Comparison of the SD retrieval results with ground-based measurements

Figure 6 displays a comparison of the station measurements and the C-snow products at different scales. As the scale increases, corr.coe decreases from 0.52 to 0.33. Additionally, ubRMSE increases from 68.18 to 77.47 cm, reflecting increased uncertainty. Moreover, as the scale increases, the mean bias transfers from a slight underestimation of -9.81 cm to a slight overestimation of 10.63 cm. The decreased correlation is reasonable because the station observations are point-scale measurements. Even for pixels with multiple stations, it is still difficult to obtain a continuously distributed SD map using a limited number of samples.



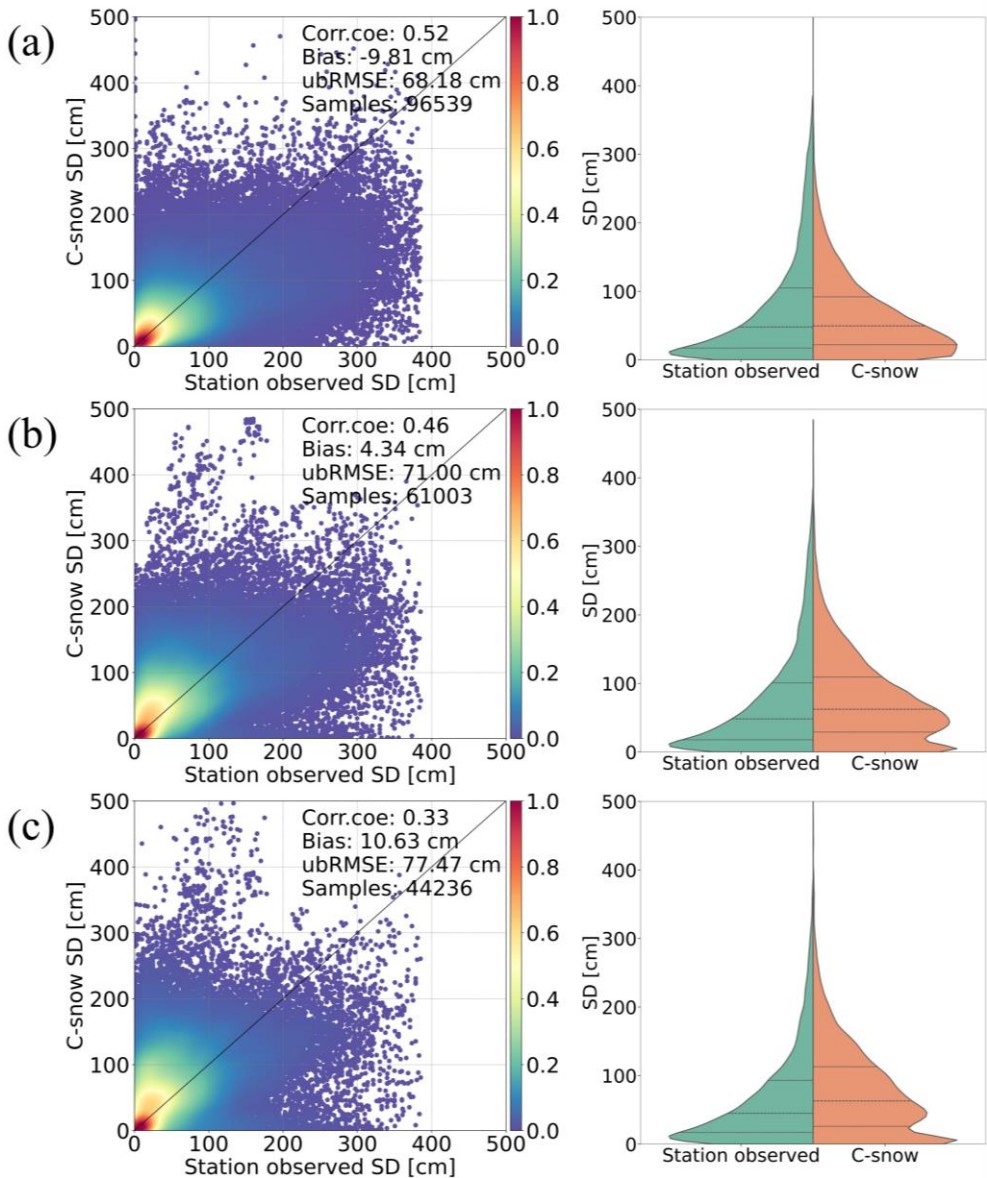

**Figure 6.** Comparisons (left column) and distributions (right column) between the C-snow SD and the station-observed SD at (a) 1-, (b) 10-, and (c) 25-km scales. The dashed lines in the right column indicate the 25th, 50th and 75th percentiles.

The spatial distributions of C-snow at different scales are displayed in Figure 7. The overall spatial patterns at the three scales are similar. However, the spatial details are different. For example, in the mountainous regions of the western United States, Europe, and Asia, the 1-km-resolution C-snow SD clearly captures more detailed features of the snowpack. In contrast, at the larger scales of 10 and 25 km, these details are masked, resulting in a loss of spatial distribution information for SD.





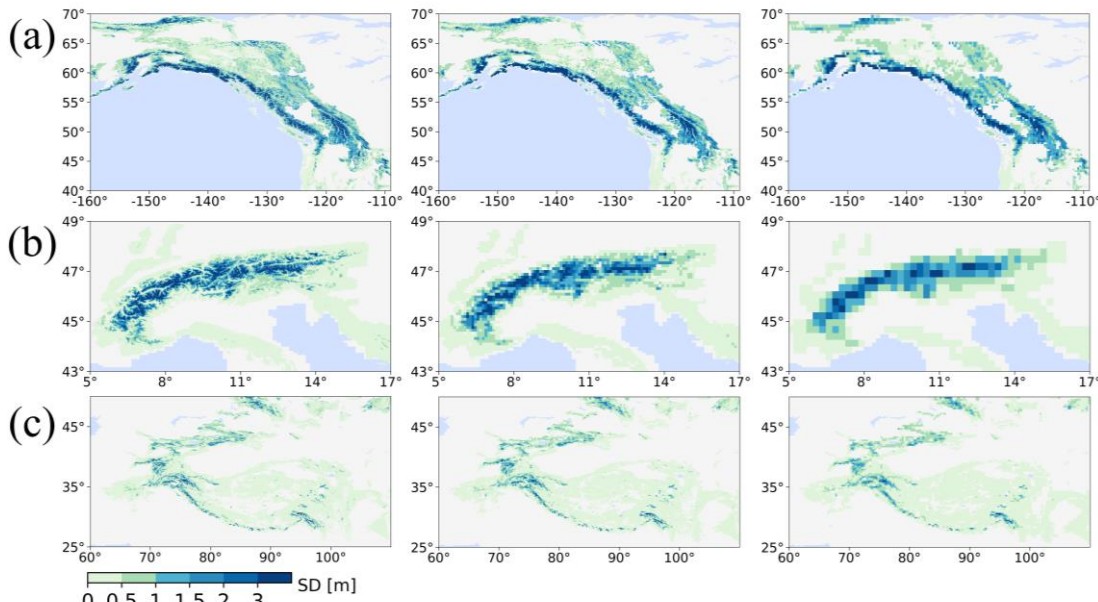

**Figure 7.** Spatial distribution of the monthly average C-snow SD in three mountainous regions in February 2018 at three scales: (a) the
western mountainous region of the United States, (b) the European Alps, and (c) the Hindu-Kush Himalayas. The left, middle and right
columns show the results at scales of 1, 10, and 25 km, respectively.

Figure 8 shows the time series of the C-snow products compared with the station measurements. In general, at all scales,

the C-snow SD is underestimated in the snowmelt season starting in March. When the scale increases from 1 to 25 km, we

observe an increase in the mismatch of snow season length between the C-snow and station SDs. In addition, the average SD

from the stations becomes increasingly greater than the C-snow SD during the dry snow season and increasingly lower

during the wet snow season. This explains the decreased correlation between the two datasets as the scale increases in terms

of temporal variation. The underestimation of the SD for C-snow at 1 km in the wet snow season can be explained as the

damping effect of liquid water on microwaves. The differences at 25 km are more difficult to explain. This is because a long

snow season is usually associated with a high SD. Therefore, this pattern makes it difficult to provide a consistent answer as

to why the stations are characterized by lower SD in the dry snow season and higher SD in the wet snow season. To explore

this question, we must assess the relevant spatially distributed influential factors.





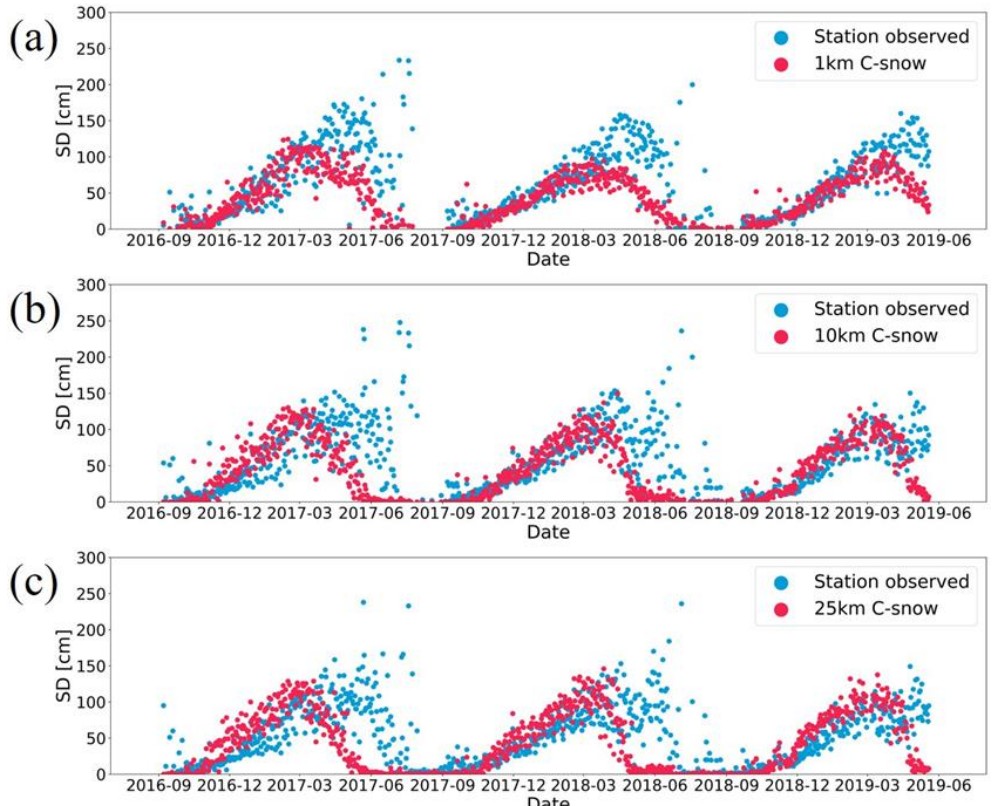

**Figure 8.** Time series of C-snow and station-observed SD at scales of (a) 1, (b) 10, and (c) 25 km.

## 3.2 Comparison of the SD retrieval results with the ASO LiDAR data

In this section, the C-snow retrieval results are assessed at different scales on the basis of the ASO data (Figure 9). At the 1 km scale, C-snow SD is underestimated, with a bias value of -91.31 cm and an ubRMSE as high as 104.3 cm. As the scale increases, the accuracy of C-snow increases, with the ubRMSE decreasing from 104.30 to 83.21 cm and the bias decreasing from -91.31 to -52.73 cm. Overall, the results of the ASO-based validation indicate an increasing trend in the accuracy of C-snow as the scale increases, which is different from the conclusions based on the station-based measurements

in Figure 6. A detailed discussion of this difference is provided in Section 4.1.





**Figure 9.** Comparisons (left column) and distributions (right column) between C-snow SD products and ASO SD at different scales, where (a), (b), and (c) represent scales of 1, 10, and 25 km, respectively. The dashed lines in the right column indicate the 25th, 50th, and 75th percentiles.

Figure 10 shows a comparison of time series of the C-snow SDs and ASO observations in different basins at three scales. The gray points represent the daily average air temperature. The red points represent the average daily C-snow SD in all the selected basins, and the different blue symbols indicate the daily average ASO data in various basins. At all scales, the C-snow retrievals match well with the ASO data in 2017. In 2018, when the ASO observations are primarily concentrated between March and June, the C-snow retrieval results are underestimated because of wet snow. During the heavy snow



season of 2019, the C-snow SD are significantly underestimated compared with the ASO data and fail to accurately capture the changes in the snowpack during this period.

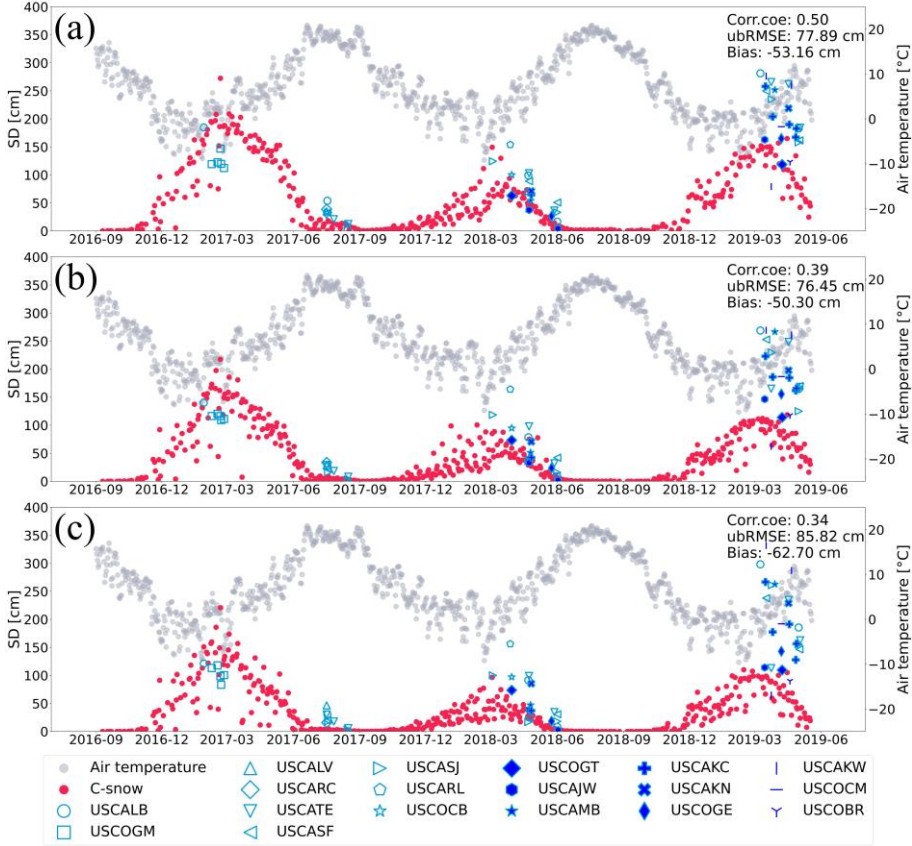

**Figure 10.** Time series comparison of C-snow products with ASO observations for various basins at (a) 1-, (b) 10-, and (c) 25 km scales.

**3.3 Effects of landscape and terrain on the SD retrieval results**

The impact of land cover on C-snow accuracy was investigated at different scales, as shown in Figure 11. Here, station measurements were used as reference data because of their global coverage. In forested covered regions, C-snow tend to be slightly underestimated in general. The accuracy decreases with increasing scale, with corr.coe decreasing from 0.52 to 0.35 and ubRMSE increasing from 69.45 to 76.07 cm. In the permanent ice region, the C-snow product includes several abnormally overestimated results, especially at the 10- and 25-km scales, with a bias over 290 cm. For other types, C-snow

also displays a decrease in accuracy at 10- and 25-km resolutions compared with that at the 1-km scale.



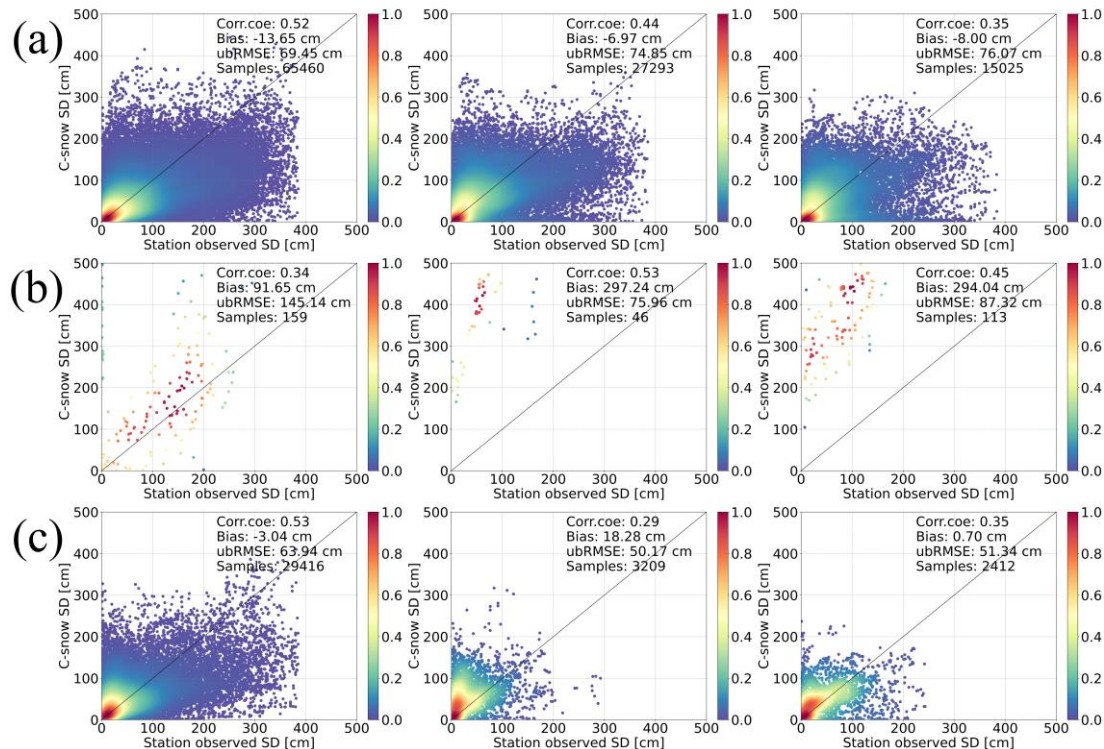

**Figure 11.** Impact of various land cover types on the accuracy of C-snow products at different scales: (a) tree cover, (b) permanent ice, and (c) other types. The left column represents the scale of 1 km, the middle column represents the scale of 10 km, and the right column represents the scale of 25 km.

Figure 12 shows the impact of the SD conditions and various geographic environments on the accuracy of C-snow data at different scales. Here, the samples with permanent ice cover were excluded from the validation data because of large errors (see Figure 11b). When the SD at a station is less than 100 cm, all three scales are overestimated (Figure 12a). The overestimation becomes more pronounced as the scale increases, with Rbias increasing from 38.44% at 1 km to 83.74%. In contrast, when the SD exceeds 100 cm, all scales show underestimation. The underestimation is relatively small at the 10 km

scale, where Rbias reaches -47.80% when the SD is greater than 200 cm. For tree cover, the results at the three scales tend to be underestimated (Figure 12b). For the other land cover types, Rbias at the 10 km scale reaches as high as 46.97%, indicating significant overestimation. When the forest fraction is between 0 and 0.2, significant overestimation occurs at both the 10- and 25-km scales, with Rbias values of 74.29% and 107.14%, respectively (Figure 12c).

        At elevations below 1000 m, the C-snow product is overestimated at the 1 km scale, with an Rbias of 26.48%, whereas

it is underestimated at all other elevation intervals (Figure 12d). For elevations between 2000 and 3000 m, the values at both the 10- and 25-km scales are underestimated, with Rbias values of -13.82% and -13.70%, respectively. When the standard deviation of elevation is less than 50 m, C-snow is significantly overestimated, with an Rbias value as high as 47.56% (Figure 12e). When the standard deviation of elevation is greater than 100 m, the C-snow values at both the 10- and 25-km scales are overestimated, with an Rbias value of up to 26.82%. We also find that C-snow performs best in areas with



moderate standard deviations of elevation (50-100) and moderately to highly forested (0.4-0.8) areas. As the elevation
difference between the station and the grid increases, an underestimation trend is observed at all scales at the 25 km scale,
with Rbias ranging from 24.16% to -60.13% (Figure 12f). When the station elevation is lower than the grid elevation
(elevation difference < 0), underestimation is only observed at the 1 km scale, with an Rbias of -25.24%.

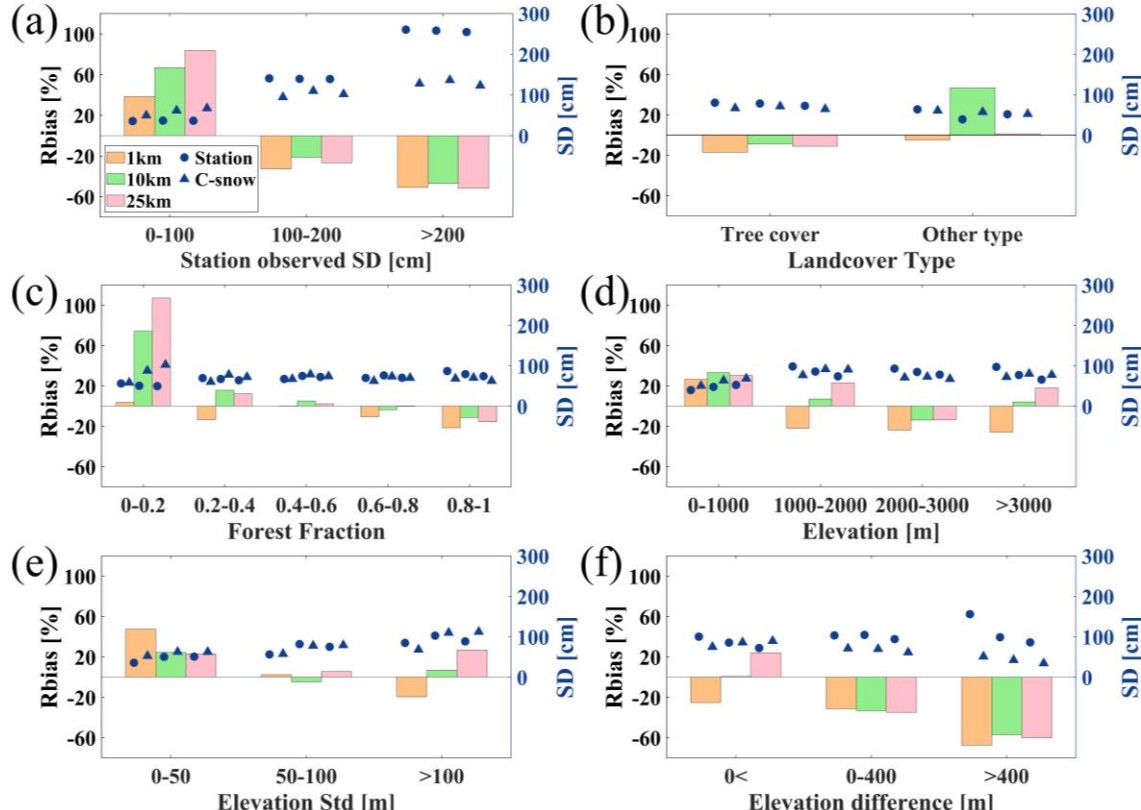

**Figure 12.** Impact of different (a) station-observed SD, (b) land cover types, (c) forest fractions, (d) elevations, (e) standard deviations of
elevation, and (f) elevation differences between stations and grids on the accuracy of C-snow SD products across various scales, where the
left axis represents Rbias and the right axis represents the average SD observed at stations and in the C-snow products.

Figure 13 shows the spatial distributions of Rbias at different scales. A total of 38.95% of the grids with Rbias values
lower than 0 are underestimated at the 1-km scale, and 34.22% of the grids with Rbias values greater than 100% are
significantly overestimated, especially in the western mountain ranges of the United States, the Appalachian Mountains, the
southern part of the Scandinavian Mountains, the European Alps, and the Hindu-Kush Himalayas. Moreover, this trend
becomes more pronounced with increasing scale, with Rbias values greater than 100% accounting for 39.94% and 39.45% of
all values at the 10- and 25 km scales, respectively.



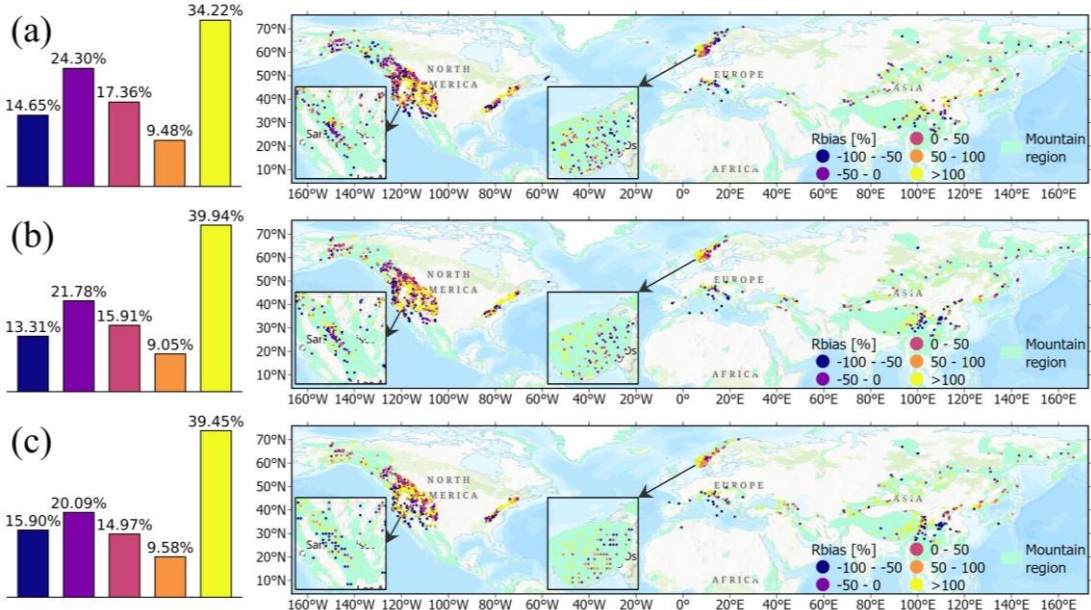

**Figure 13.** Histograms (the left column) and spatial distributions (the right column) of Rbias at (a) 1-, (b) 10-, and (c) 25 km scales.

### 3.4 Influence of complex geographic environments on SD retrieval

To explore the influence of complex geography on C-snow retrieval, we selected three nested grids of C-snow retrieval results at different scales (1, 10, and 25 km) and the corresponding station observation data. Figure 14 displays the discrepancies in geographic environments among these nested grids. Within the first and third nested grids, there is only one station, corresponding to the three scale grids. In the second nested grid, there are four stations, which correspond to the four 1 km C-snow grids and three 10 km C-snow grids. The average values are calculated for the C-snow product and the station observations.

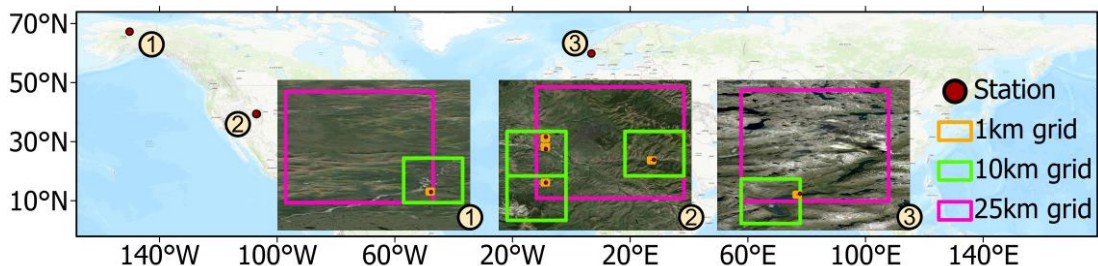

**Figure 14.** Spatial locations of the three selected nested grids, with the first at 67.45°N, -150.31°E; the second at 39.42°N, -107.00°E; and the third at 56.00°N, 6.85°E.

In the first nested grid, the station observations generally match the C-snow retrieval results at three scales (Figure 15a). In the second nested grid, the station-observed snow cover is quite shallow, whereas the C-snow values at three scales are overestimated relative to the station observations, especially during the period from December 2016 to March 2017 (Figure




15b). In the third nested grid, the time series changes at the 1 km scale for C-snow closely match those of the station
observations, whereas the C-snow retrieval results are overestimated relative to the station observations at both the 10- and
25 km scales (Figure 15c). Moreover, we find a large discrepancy between the C-snow at the 10- and 25-km scales.

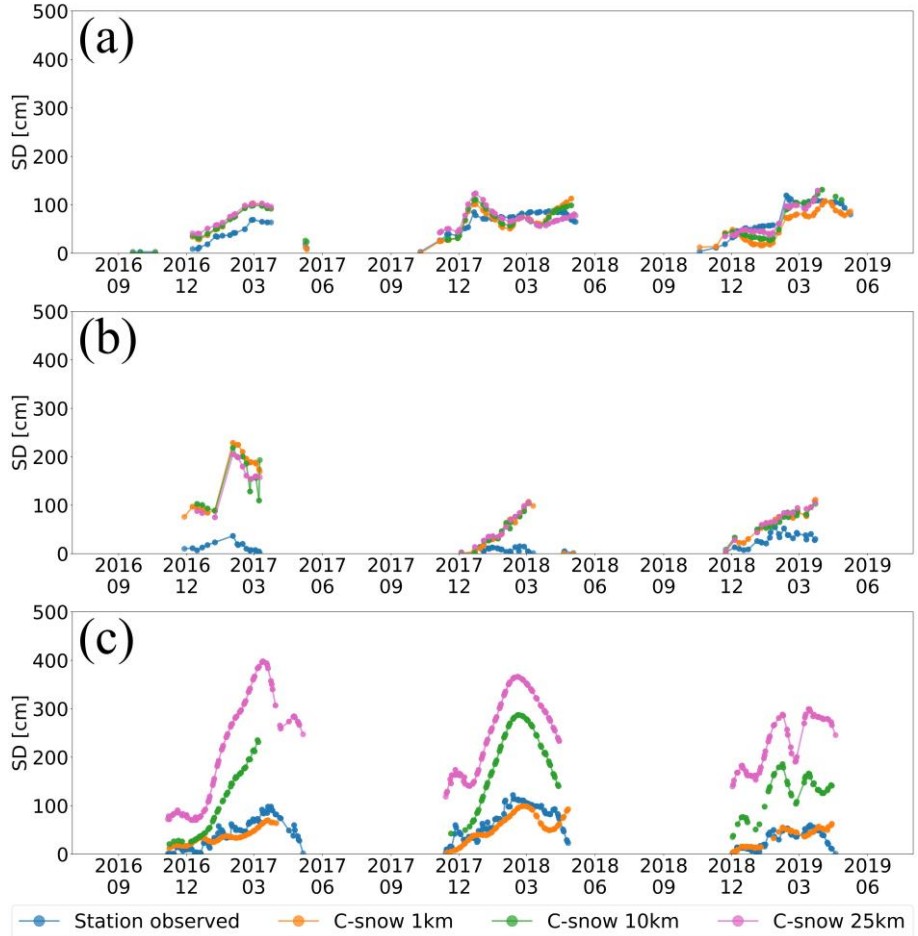

**Figure 15.** Time series of SD at different scales (1, 10, and 25 km) for three selected nested grids, where (a), (b), and (c) represent the
results for the first, second, and third grids, respectively.

To explain the scale effects associated with geographical heterogeneity, we calculated the differences between the 10-
and 25-km grids in terms of land cover type, forest fraction, elevation, and standard deviation of elevation (Figure 16). For
the first nested grid, the terrain is relatively flat (mainly below 1000 m) at both the 10- and 25-km scales, which ensures the
representativeness of the station to a certain extent. Moreover, the snowpack is below 100 cm where the C-snow retrievals
perform well (Figure 6 and Figure 8). Thus, the C-snow retrievals at 1, 10 and 25 km are in good agreement with the station
observations. For the second nested grid, the geographic environments at the 10- and 25-km scales are very similar; thus, the
SD retrievals at both scales are similar. However, the terrain is very complex, e.g., high elevation (2000-3000 m) and high
topographic relief (0-300). Thus, the representativeness of the stations may be problematic, resulting in poor agreement with





the C-snow retrieval results. For the third nested grid, we find that the heterogeneity of the 10- and 25-km grids is high. For example, the coverage of permanent ice (24.96%) at the 25 km scale is high relative to that at the 10 km scale, whereas the tree cover fraction (25%) in the 10 km grid is high. Additionally, the terrain is more complex in the 25 km grid than in the 10 km grid, e.g., at high altitudes. The overestimation occurs mainly because of permanent ice, which is consistent with the results in Figure 11. The large differences (scale effects) in the SD retrieval results at the 10- and 25-km scales are related to the heterogeneity of the geographic environment. Specifically, the greater the heterogeneity of the geographic environment between the 10- and 25-km scales is, the greater the differences in the SD retrievals.

**Figure 16.** Distributions of (a) land cover types, (b) forest fraction, (c) elevation, and (d) standard deviation of elevation in three selected nested grids, with the left, middle, and right columns representing the first, middle, and third grids, respectively.





## 4 Discussion

### 4.1 Different scale patterns of C-snow retrievals with station and ASO measurements

In this study, we compared the C-snow retrievals with both station data and ASO observations across various scales, with different trends identified with increasing scale. Compared with that of the station data, the accuracy of the C-snow data tended to decrease as the scale increased, whereas the accuracy of the ASO data tended to increase (Figure 17a). To explain this inconsistency, we further compared the station data and ASO observations (Figure 17b). There is a significant correlation between the station and ASO data, with a corr.coe of 0.79. Moreover, the bias is -31.60 cm, and the ubRMSE is

41.19 cm, indicating that some errors remain. Thus, the uncertainty of the ASO data may affect the results, although the data are reliable. The ASO SD is calculated using scanning LiDAR measurements, a straightforward and robust approach involving the subtraction of snow-free surface elevation data from snowpack surface elevation data. The accuracy of LiDAR-derived SD is affected by factors such as terrain and vegetation cover (Enderlin et al., 2022; Neuenschwander et al., 2020; Klápště et al., 2020). Within the coverage scope of ASO data, steep slopes (as high as 80 degrees) and high forest

fractions (mean value of 53%) likely affect the accuracy of the observations (Figure A1).

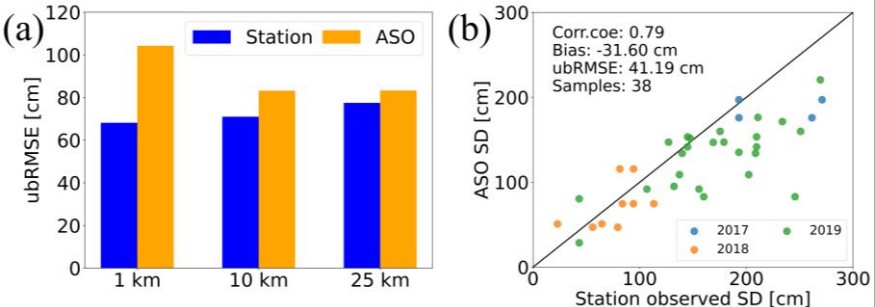

**Figure 17.** (a) Accuracy performance of the C-snow product at different scales when station observations and ASO SD data are used as reference data and (b) a comparison of station data and ASO observations (3-m ASO data are used to match the station data).

      Another reason for the contrasting accuracy trend is the variation in the representativeness of the SD stations. To

address this issue, we counted the number of stations within each grid at the 10- and 25-km scales (Figure A2). We found that most grids, namely, 83.54% at the 10 km scale and 62.12% at the 25 km scale, contain only one station. We further compared the accuracy of the C-snow retrievals from grids with only one station and those with more than one station (Figure 18). The results show that the performance of C-snow is related to the number of stations in the sample grids. For example, at the 10 km scale, corr.coe increases from 0.44 to 0.55 and ubRMSE decreases from 71.63 to 66.07 cm with

increasing number of stations. At the 25 km scale, the improvement in accuracy is not obvious, with the corr.coe improving from 0.33 to 0.35 and the ubRMSE decreasing from 77.11 to 75.86 cm. This is reasonable because the coarser the grids are, the more stations need to ensure the representativeness of the true data. In addition, the C-snow retrievals from grids with only one station are overestimated, and the bias ranges from 7.12 to 20.44 cm. For grids with more than one station, the C-snow retrievals are typically underestimated, with bias ranging from -3.19 to -9.45 cm. Thus, the representativeness of




stations at large scales, such as 10 and 25 km, is very important for achieving reasonable validation results. Beyond the representativeness of the stations, the method used to convert point-scale observations to the spatial scale of satellite pixels also affects the validation results (Fassnacht et al., 2006; Hou et al., 2022), this study employed a simple averaging method, which somewhat ignores spatial variability (Ge et al., 2019).

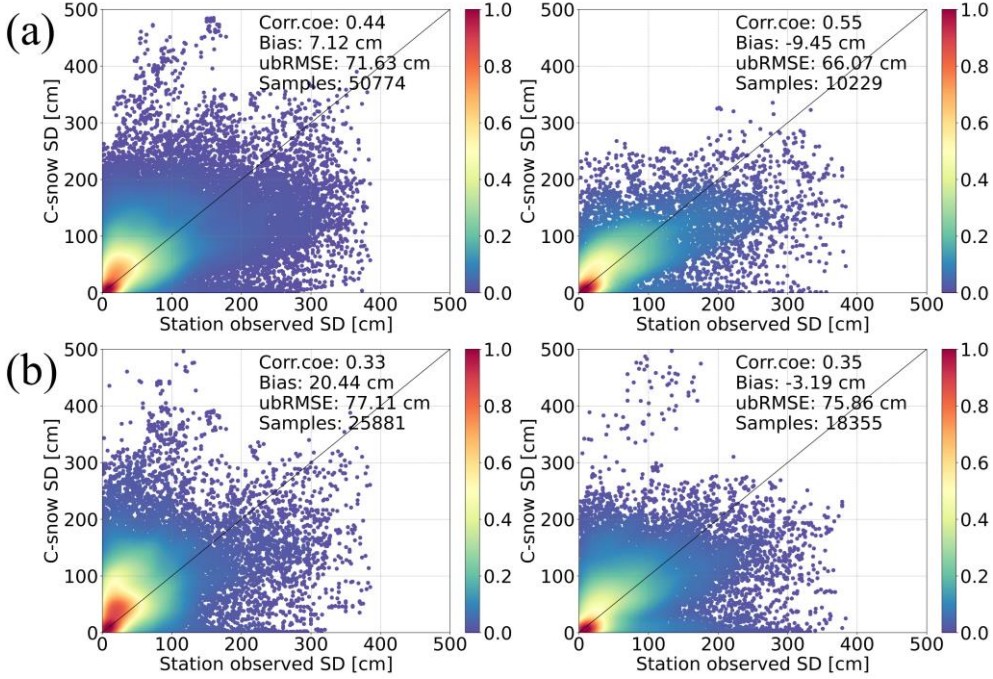

**Figure 18.** The impact of the number of stations within grids at the (a) 10- and (b) 25-km scales on the accuracy of the C-snow product. The left column presents the C-snow evaluation result when there is only one station within the grid, and the right column presents the evaluation result when there is more than one station within the grid.

### 4.2 Overestimation in permanent ice landscapes

   The presence of permanent ice results in large errors in the accuracy of C-snow at different scales (Figure 11). We further analyzed the errors in the C-snow product as the coverage of permanent ice within the grids increased at the 10- and 25-km scales (Figure 19). With increasing permanent ice coverage, the bias gradually increases at both the 10- and 25-km scales, clearly indicating an overestimation trend (Figure 19a). Moreover, the ubRMSE also presents an increasing trend, from 67.00 to 214.24 cm, indicating high uncertainty due to the presence of permanent ice (Figure 19b).





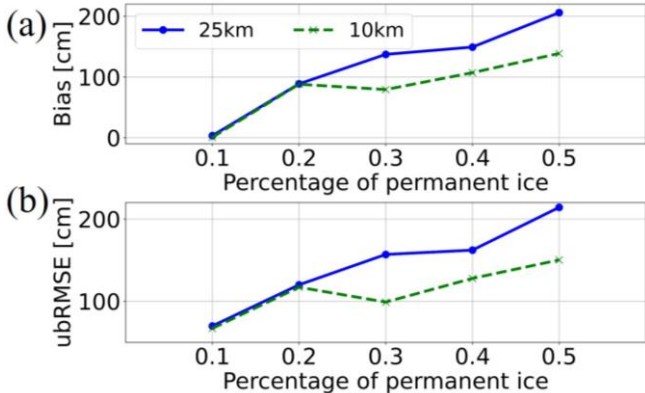

**Figure 19.** Impact of permanent ice on the accuracy of the C-snow product at different scales, including (a) bias and (b) ubRMSE, which are statistics based on the station dataset.

We also selected three nested grids to show the time series SD data at three scales (Figure A3). In the first grid, there is no ice present in the 1 km grid. In the 10 km grid, the ice coverage reaches 29.30%, whereas it is as high as 62.42% within the 25 km grid. In the second nested grid, the percentages of ice coverage are 11.60% and 61.31% for the 10- and 25 km grids, respectively. In the third nested grid, the ice coverage within the 10 km grid (68.0%) exceeds that within the 25 km grid (48.26%).

The time series of C-snow retrieval results within three selected nested grids is shown in Figure A4, together with the station data at different scales (1, 10, and 25 km). Notably, the percentage of ice coverage dominates the bias in the C-snow retrievals in both the 10- and 25-km grids. For example, the C-snow retrieval results at the 25 km scale are significantly overestimated compared with the station observations because of the high degree of ice coverage (Figure A4a, Figure A4b). The C-snow retrievals at the 10 km scale are seriously overestimated compared with those at the 25 km scale because of the high ice coverage (Figure A3). Permanent ice exhibits similar electromagnetic properties to those of snowpacks, enhancing the backscattering of radar signals (Scott et al., 2006). During the melt season, an increase in the roughness of the ice surface leads to an increase in the backscattering coefficient (Baumgartner et al., 1999). The dynamic nature of glaciers, characterized by crevasses and glacier movement, can lead to temporal variations in the backscattering coefficient (Sander and Bickel, 2007; Brock and Billy, 2010), complicating interactions between radar signals and snow characterization. Thus, quality control of spatially sampled C-snow products, especially at coarse scales, must be performed, ensuring that the retrieval results in permanent ice-covered areas are filtered and removed.

## 5 Conclusion

In this study, we evaluated and compared the accuracies of C-snow retrieval results at three spatial scales (1, 10, and 25 km) through station measurements and ASO observations. We also analyzed the factors influencing the accuracy at these scales and explored the inconsistency in scale effects via station and airborne reference datasets. Our results indicate that as the





spatial scale increases, the correlation between the C-snow products and station observations significantly decreases, with a corr.coe of 0.52 at the 1 km scale, which decreases to 0.46 at the 10 km scale and 0.33 at the 25 km scale. The error increases

with scale, from 68.18 cm at the 1 km scale to 77.47 cm at the 25 km scale. Compared with the airborne ASO data, the C-snow product became increasingly more accurate as the spatial scale increased. with bias values ranging from -91.31 to -52.73 cm and ubRMSE values ranging from 104.3 to 83.29 cm. These different scale patterns occur mainly because of the different representativeness of station and ASO data.

We also found that the land cover type affects the accuracy of C-snow classification. In areas covered with tree cover, the

accuracy of C-snow significantly decreases as the spatial scale increases, with the corr.coe decreasing from 0.52 to 0.35 and the ubRMSE increasing from 69.45 to 76.07 cm. In areas covered by permanent ice, C-snow consistently overestimates SD at all scales, which is related to the percentage of ice coverage. The impact of terrain on the accuracy of C-snow is complex. The overestimation of C-snow at the 1 km scale is evident for elevations below 1000 m, whereas SD tends to be underestimated in other elevation ranges. For elevations between 2000 and 3000 m, the C-snow retrievals at both the 10- and

25-km scales displayed an underestimation trend, with Rbias values of -13.82% and -13.70%, respectively. The standard deviation of elevation also affects the accuracy of C-snow. When the standard deviation of elevation is less than 50 m, C-snow at the 1 km scale is overestimated (Rbias of 47.56%), and when the standard deviation of elevation is greater than 100 m, C-snow at the 25 km scale is overestimated (Rbias of 26.82%).

In this study, we assessed the performance of C-snow products at different spatial scales and analized the corresponding

influencing factors. According to our study, C-snow products at the three scales are characterized by high uncertainty. Particularly, we should be careful when using coarse-scale C-snow products as a reference, and at a minimum, some outlier data should be filtered and removed. Future research should continue to explore the possibility of improving C-snow retrieval by combining SAR backscattering, polarimetric, interferometric and satellite LiDAR data to enhance the reliability and accuracy of Sentinel-1-based products in practical applications.




**Appendix**

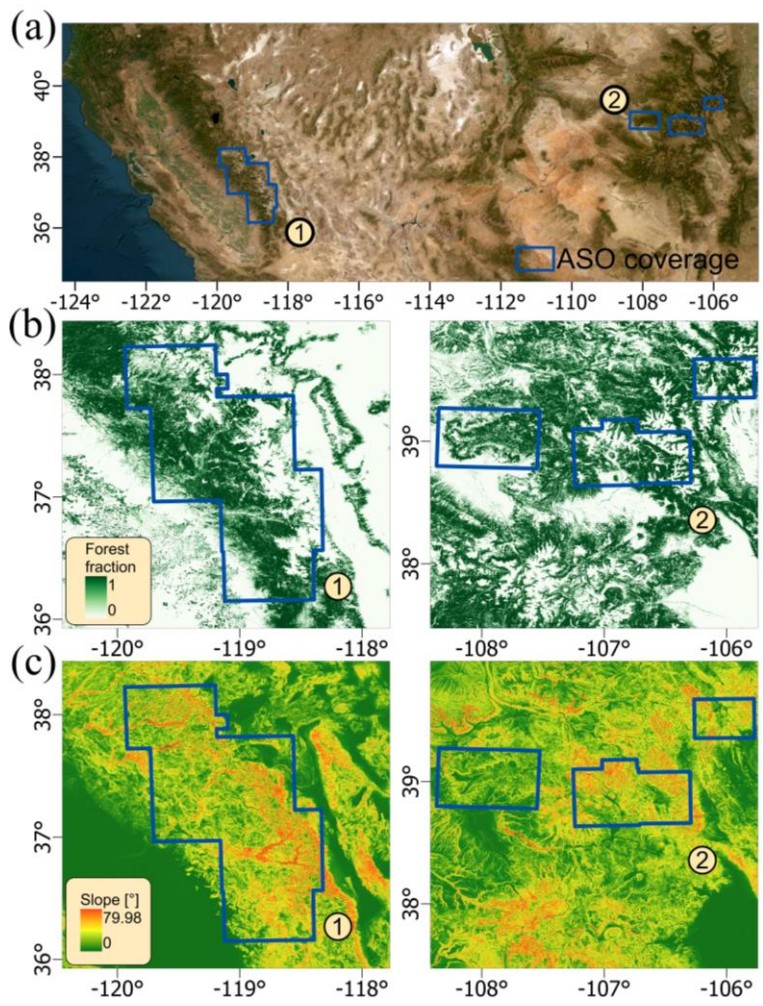

**Figure A1.** (a) The overall geographical conditions within the coverage area of ASO, with zoomed-in views of (b) the forest fraction and (c) slope conditions.

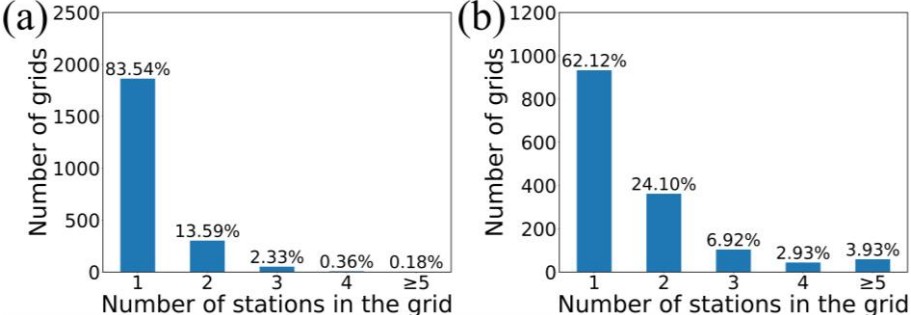

**Figure A2.** Statistics regarding the number of stations within the grids at scales of (a) 10 km and (b) 25 km.



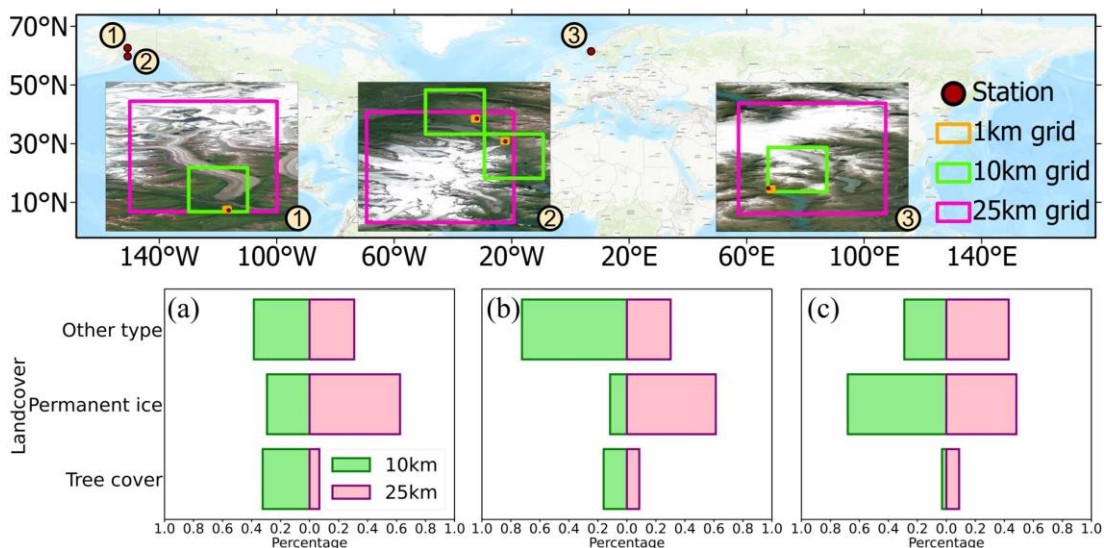

**Figure A3.** Spatial distributions and summaries of tree cover, permanent ice, and other coverage types in three selected nested grids: (a) the first 25 km grid at 62.84°N, -150.82°E, (b) the second 25 km grid at 59.61°N, -150.82°E, and (c) the third 25 km grid at 61.59°N, 7.11°E.

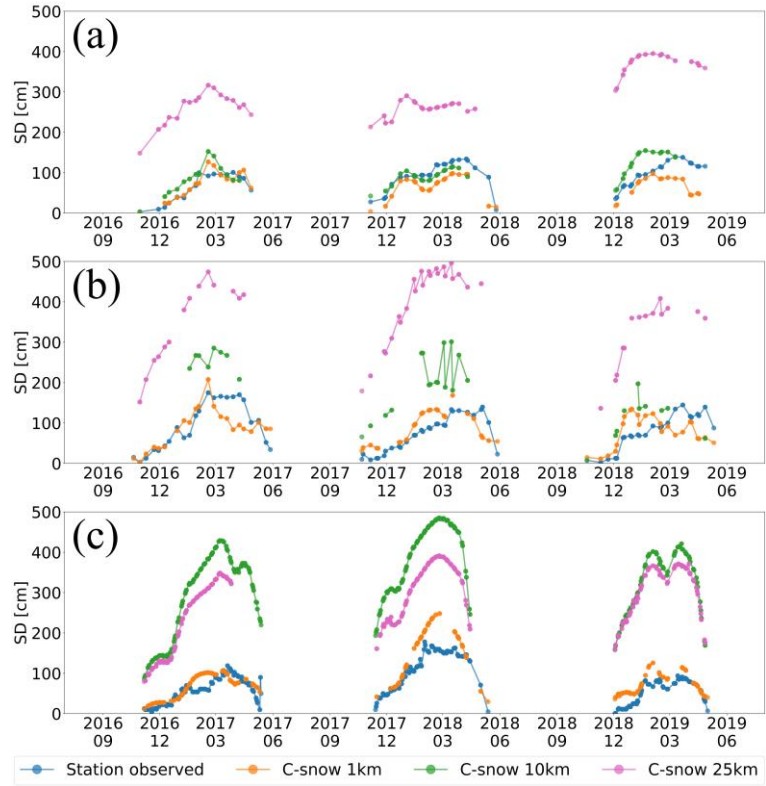


**Figure A4.** Time series of the C-snow retrieval results and station measurements at three scales (1, 10, and 25 km) within the three specifically selected nested grids in Figure A3.



*Code availability.* The code is available by contacting the corresponding author (yangjw@bnu.edu.cn).


*Author contributions.* JJY carried out the analyses, created the figures, and wrote the manuscript. JWY helped design the research, write the manuscript and edit manuscript text. LMJ, JMP, CX helped in editing the manuscript.

*Competing interests.* The contact author has declared that none of the authors has any competing interests.

*Acknowledgments.* This work was supported by the National Natural Science Foundation of China (42201346, 42090014, and 42171317) and the National Key Research and Development Program of China (2021YFB3900104).

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
