# Peer review of "Scale patterns of the Sentinel-1 SAR-based snow depth product compared with station measurements and airborne LiDAR observations"

_EGUsphere, 2025_

## Referee Comment (RC1)

Review of the manuscript egusphere-2025-276 entitled "Scale patterns of the Sentinel-1 SAR-based snow depth product compared to station measurements and airborne LiDAR observations" by Ying et al.

This article conducts a thorough evaluation of the multi-scale performance of the Sentinel-1 SAR snow depth product, offering notable value in data validation and environmental impact analysis, with clear figures and fluent expression. However, its focus on validating existing algorithms rather than achieving a breakthrough limits its innovation, and certain language expressions lack smoothness. It is recommended that the manuscript be revised prior to submission for publication.

**Major**

1. Although the article systematically evaluates the C-snow product across multiple scales, its core methodology—such as the C-band SAR-based snow depth retrieval algorithm—is not original to this study but builds upon prior work by Lievens et al. (2019). The innovation here lies primarily in data validation and scale analysis, yet these aspects do not represent a novel breakthrough in the field of remote sensing. It is recommended that the study explicitly highlight its unique contributions, such as whether it proposes a new scale-effect model or an improved retrieval method.

2. The introduction provides a detailed review of the development of SAR and microwave remote sensing in snow depth monitoring but fails to adequately justify the selection of the 1, 10, and 25 km scales for analysis or clarify their relevance to practical applications, such as hydrological modeling.

**minor**

1. Lines 20-21: Sentence is too long. "The results indicate that the scale patterns of the C-snow products across various resolutions differ from those of station- and airborne-based reference data." → "The scale patterns of C-snow products vary across resolutions. They differ from patterns observed in station and airborne reference data."

2. Lines 366-367: The text contains redundant expressions, and optimization is recommended. "bias values ranging from -91.31 to -52.73 cm and ubRMSE values ranging from 104.3 to 83.29 cm"→"bias values decrease from -91.31 cm to -52.73 cm, while ubRMSE decreases from 104.3 cm to 83.29 cm".

3. Line 379: analized → analyzed.

4. Suggestions for unified terminology. "ground-based measurements", "station observations".

---

## Referee Comment (RC2)

**Scale patterns of the Sentinel-1 SAR-based snow depth product compared to station measurements and airborne LiDAR observations**

**General Comments:**

This paper addresses an important topic by evaluating the scale-dependent performance of the C-SNOW Sentinel-1 snow depth product against both in-situ measurements and airborne LiDAR observations. The multi-scale analysis and inclusion of geographic and land cover effects are valuable contributions to the remote sensing and snow hydrology communities. However, the manuscript would benefit from greater clarity in its introduction and methodological explanations, Additionally, some of the comparisons between datasets are not well-aligned in terms of spatial or temporal scale, which limits the interpretability of the findings. The Discussion section leans heavily on restating results rather than offering critical insight into the causes and implications of observed discrepancies.

Overall, the paper has the potential to contribute knowledge on the accuracy of Sentinel-1 snow depth at different spatial scales, but revisions are needed to improve its structure, clarity, precision, scientific depth, and accuracy.

**Major Comments:**

1. The rationale behind the use of Sentinel-1 data at 10 km and 25 km resolutions is unclear. If the primary aim of the paper is to evaluate the spatial and temporal performance of Sentinel-1 for snow monitoring, then standard higher-resolution products (e.g., 0.5 km and 1 km) would suffice. The inclusion of coarser resolutions needs to be better motivated and should be clearly stated and supported by appropriate literature and methodological context.

2. Since NRCS SNOTEL provides direct snow depth measurements, please clarify why a conversion from SWE to SD was performed. A fixed density to go from SWE to depth will impact the S1 evaluation, which is another consideration. Also, explicitly explain the choice of using a fixed snow density value of 0.24 g/cm³ for the Russia-SWE dataset. Is this based on regional averages or prior literature? Including a brief rationale would improve clarity.

3. In Figure 10, the comparison of the C-SNOW time series with spatially distributed ASO LiDAR data appears unrelated comparison. Since C-SNOW is also available as a spatial product over much of the Northern Hemisphere, it would be more appropriate to extract the C-SNOW spatial data closest to date to the ASO flight and perform a spatially explicit comparison. This would enable more meaningful evaluation and avoid misleading conclusions from mixed-scale comparison. Furthermore, the inclusion of air temperature in this figure is puzzling. The manuscript does not provide a rationale in the methodology for using temperature as a covariate or validation proxy. Since it is not used quantitatively in the analysis, I recommend removing it unless a clear scientific justification is provided. Overall, please revise this section to focus on meaningful spatial comparisons (e.g., using representative basins), and consider summarizing the evaluation using average statistics or appropriate spatial accuracy plots (scatter, bias maps, etc.).

4. There are several issues in section 3.3. First, the phrase "other types" of land cover should be clarified—please specify which land cover categories are included beyond forest and permanent ice. Additionally, while the inclusion of permanent ice regions is noted, it's important to question the relevance of evaluating C-snow accuracy in such areas. Does the Lievens et al. algorithm or Sentinel-1 backscatter perform reliably in permanent ice environments and dense forests? Please justify why this analysis was included and consider whether they should be treated as known limitations of the remote sensing platform and dataset. In continuation, some findings are counterintuitive and warrant further explanation. For instance, significant overestimation in areas with low forest fraction (0–0.2, figure 12c) is unexpected, as reduced vegetation cover typically enhances radar retrieval accuracy. Similarly, while elevation and elevation variability appear to influence performance, the paragraph lacks synthesis on why certain ranges (e.g., moderate forest and elevation variability) yield better results. Please also clarify how elevation differences between stations and grid cells were calculated, and how these mismatches propagate error across scales (this can go in methodology).

5. The current Discussion section is written more in the style of a results narrative. While the detailed reporting of accuracy metrics, grid-level behavior, and ice-related overestimates is important, most of the text focuses on what was found rather than interpreting what it means. To strengthen this section, I recommend separating the descriptive content into the

results section and expanding the discussion with deeper analysis. For example, lay error analysis background, consider critically why ASO and station-based validation trends differ, what the implications of spatial representativeness are for coarse-scale validation, and how the known limitations of Sentinel-1 in forested or glaciated areas affect the broader applicability of C-snow. Comparisons with previous studies and a more explicit articulation of limitations and future directions would also help to better contextualize the results.

**Minor Comments**

1. The introduction is currently in a general tone. Several statements are made without citing relevant studies or offering clear justification. Please consider grounding key claims with references and clarifying the motivation and novelty of the study more explicitly.

2. Land cover classifications such as forest and permanent ice are introduced in the Results section without prior explanation. These should be defined and justified in the Methodology section, including the source of the land cover data and how the categories were used in the analysis.

3. Sections 3.3 and 3.4 both focus on geographic and environmental influences on snow depth retrieval. Merging them into a single cohesive section would improve readability and thematic consistency by presenting all location-based findings together.

4. Reorganize Figures and reconsider adding all figures for Improved Flow
   a. Figure 14 could be merged with Figure 2 to streamline the presentation of the study area and grid setup. Label the nested grid structure directly in the map and refer to it when introducing the experimental design.
   b. Figure 16 is conceptually related to Figure 12. Placing them closer together would enhance the narrative flow and allow readers to better understand the progression of results.
   c. Please consider reducing the number of figures and balancing the proportion of figures and text. Refer to comments that suggest either removing or combining figures.

**Line-to-Line Comments**

**23 and elsewhere:** The errors statistics are reported to the hundredth of a cm, but this level or precision is not realistic or warranted. Please consider the significant units here and elsewhere when reporting errors.

**26:** Remove "Especially an".

**35:** To clarify how snow depth (SD) information contributes to water availability, state "…estimated from snow depth (SD) and snow density.".

**44:** Add "one or several" before "meters".

**45:** These density values are too high for most seasonal snow and in the range of when snow transitions to firn. Recommend revising to the typical range of density values (~100 to 550 kg/m3) for seasonal snow (see Sturm et al., 2010, J. Hydrometeorology).

**38- 50:** The paragraph introduces the use of microwave remote sensing for SWE retrieval and suggests that SAR offers advantages over passive microwave techniques. However, it would benefit from greater specificity and clarity. For instance, clearly states that while passive microwave remote sensing is widely used, its coarse spatial resolution (~25 km) limits its ability to capture the fine-scale spatiotemporal variability of snowpacks in complex mountainous terrain. This will help establish a stronger context for the discussion of SAR advantages.

Additionally, when discussing mountain snowpack complexities (**lines 45–50**), it would be helpful to explicitly connect these challenges—such as variable snow density, grain size, wind, and gravity-driven compaction, elevation, and aspect—to the limitations of passive microwave sensing. You may also consider briefly noting the difficulty of ground-based observations in remote, high-elevation regions, which further highlights the value of satellite-based approaches. These additions will create a smoother transition to the next paragraph (**lines 51–61**), which focuses on SAR. Lastly, in line 43, replace "snow cover" with "snowpack" for technical accuracy.

**51:** Delete "the" before "monitoring".

**51-52:** While this paragraph highlights recent advances using C-band SAR for SWE monitoring, it would be helpful to first acknowledge earlier foundational studies that have explored SAR for retrieving snow characteristics such as (Ulaby and Stiles, 1980; Bernier et al., 1999; Shi and Dozier, 2000; Chang et al., 2014; Lievens et al., 2019), there are many more. Including a broader

set of references would better reflect the extensive work by the remote sensing community and provide context for the transition to C-band applications.

**52-54:** The sentence discussing snow volume scattering being stronger in Ku-band compared to "other bands" could be made clearer. Please specify which bands are being referenced (e.g., X-, C-, or L-band) and provide clearer citations. If this theoretical point is derived from Rott et al. (2010) or others, please state this explicitly and consider rephrasing for clarity and precision.

**55-56:** Instead of using "VH/VV," which may be unclear to some readers, consider using the more descriptive phrase "cross- to co-polarization ratio."

**56-59:** To avoid redundancy, consider replacing the second use of "notably" with an alternative phrase. Additionally, rephrase "due to the nonspherical properties of snowpack" to "due to the anisotropic nature of snow grains," which is a more accurate physical description. You may also cite relevant literature on this point—Lievens et al. (2022) references several useful studies that could support this claim.

**62-64:** It would be helpful to clarify that the original C-snow product developed by Lievens et al. (2019) was produced at 1 km resolution without wet snow masking. In later studies, higher-resolution (e.g., 500 m) products were developed with wet snow flagging. Mention explicitly that there are no C-Band products at 10 km and 25 km resolution. Additionally, before introducing subsequent evaluation studies, please report the original study's metrics (e.g., RMSE, bias) from the original Lievens et al. study. This will help establish a clear baseline and better contextualize the results from later evaluations, thereby strengthening the motivation for your analysis.

**67-68:** Please clarify that Hoppinen et al. (2024) evaluated the performance of the C-snow retrieval algorithm using airborne LiDAR data across two separate water years: 2020 and 2021. As currently written, "2020–2021" may be misinterpreted as a single water year.

**69-71:** The statement about Broxton et al. (2024) and Yang et al. (2024) using C-snow products at 10 and 25-km resolutions appears to be inaccurate. Broxton et al. used machine learning to enhance C-snow snow depth estimates at 0.5 and 1 km resolutions and compared these to University of Arizona SWE and airborne LiDAR data. They did not use C-snow to improve passive microwave SWE. Additionally, the studies you cited used either the 1 km or 500m C-snow product—not the 10 or 25 km. Please check for accuracy and revise.

**72-73:** Please rephrase the sentence about Lievens et al. (2022) for clarity and precision. They used Sentinel-1 backscatter observations to retrieve snow depth across multiple resolutions in the European Alps and evaluated retrieval performance. This distinction is important to avoid confusion between backscatter versus the derived SD observation also in the monitoring and evaluating SD at different resolutions.

**84-85:** ASO data are spatially extensive than stations, but do not have wider global coverage (i.e., they are only available in the western U.S.). Please revise phrasing for accuracy.

**90-91:** Recommend using a citation such as "The first 1 km SD product based on C-band SAR, covering all mountain ranges in the Northern Hemisphere, was developed by Lievens et al. (2019). The dataset is publicly available through the C-SNOW project (C-SNOW, 2024)." Instead of using a link in the text.

**104-105:** The Zenodo link should be replaced with a proper DOI citation. Also, revise the sentence for typos and formatting—for instance, the link includes a fragment ("#YdYE...") that should be removed.

**106-112:** Remove links and include proper citations.

**127-133:** ASO is not a LiDAR mission, ASO is a company that conducts LiDAR flight surveys using an airborne laser scanner (ALS). The dataset is available from 2013-2019 and 2022 to present. Please check the 2.2.2 section for accuracy. Refer to NSIDC, ASO website, and Painter et al 2016 paper. Also, clarify the reasoning behind using 3m instead of 50m.

**130-134:** Please clarify that only California and Colorado ASO surveys were used.

**141-142:** Replace the word "corresponding datasets" with a direct reference to Table 1 for clarity. Also, in line with earlier comments, avoid including direct links (e.g., to Google Earth Engine) in the main text.

**164:** Clarify whether this is the Pearson correlation coefficient or other.

**169-170:** Add a line that explains what this section is about. It starts suddenly without making any coherence.

**178-182:** This text (and Figure 7) does not add much to the study, as it is all obvious and expected behavior for considering different spatial scales of a dataset. I recommend removing this.

**189-192:** The term "wet snow season" should be replaced with "ablation" or "melt season" to better reflect the physical processes and limitations of Sentinel-1 during this period. The phrase "mismatch of snow season length" is vague. It's not clear whether the authors refer to differences in snow onset and melt timing, the overall duration of snow cover, or specific discrepancies, C-Band data is only available till the end of April anyway as Sentinle-1 is not reliable in ablation season.

**195-196:** This line should be part of the limitation section

**223-225:** What is the other type of land cover you are talking about here? Please explain how you expect Sentinel-1 to work in permanent ice regions, what are those regions?

**271-276:** Here and elsewhere – what is the purpose of comparing a 10-km or 25-km estimate of snow depth with a station? One would not expect the station to match those very different scales.

**282-283:** What evidence do you have for this statement? I do not think that spatial representativeness is guaranteed for a station in a flat area. Additionally, I would not conflate low elevation with flat, as there can be topographic complexity/variability even at lower elevations.

**306-307:** It should be part of the data section

**Figures and Table Comments**

**Figure 1:** Use distinct colors for SNOTEL and GHCN since some of the stations are close to each other and it is hard to distinguish.

**Figures 1 and 2:** Consider combining these into a single figure with 4 panels (a-d).

**Figure 2:** Continent labels are hiding in some places. Make it consistent—no labels or labels everywhere.

**Figure 3:** This figure may not be necessary, especially given the already high number of figures (19). However, if you choose to include it, consider showing ASO data coverage across the entire US and highlighting (e.g., with a box) the region used in your analysis. This would provide helpful context without redundancy.

**Figure 4:** Consider using station codes instead of full names to reduce clutter. Add borders to markers for better visibility and clarify the meaning of each color in the figure description.

**Figure 5:** Please distinguish two boxes by panels for example say panel a and panel and state it in the figure description. Also, the arrow between accuracy analysis and uncertainty analysis is misleading. Are you trying to say uncertainty analysis comes after accuracy analysis? Regardless arrow is not needed.

**Figure 7:** It appears that Figure 7 is only showing C-snow data at different spatial scales (1, 10, and 25 km) across three mountain regions, without any comparison to reference or evaluation datasets. If that's the case, the purpose of this figure needs to be clearly stated—whether it is to demonstrate spatial variability or resolution effects. As currently placed, the figure does not align well with the section, which focuses on comparing C-snow with observed SD. Consider either relocating the figure to a more appropriate section or removing it if it doesn't directly contribute to the evaluation narrative.

**Figure 8:** Please clarify whether Figure 8 shows the average time series across all stations or if it represents individual station values. If it's an average, this should be explicitly stated in the caption and main text. However, averaging across diverse stations may mask site-specific dynamics and variability. Consider instead selecting one or a few representative stations to illustrate the temporal mismatch at different scales more clearly. Additionally, using a line plot instead of scatter points would improve readability and better highlight trends over time.

**Figure 10:** Check major comment section

**Figure 12:** It's not immediately obvious that the markers show actual SD values, and the bars show relative error. Adding a brief formula or explanation for Rbias in the caption or methods would also help.

**Figure 13:** The left plots may be better conveyed as pie charts rather than bar charts because they add up to 100%.

**Figure 14:** Refer to minor comments. Additionally, the layout of this figure is not ideal, as the three selected regions overlap the global map but instead could be shown separately with more detail.

**Figure 15:** How figure 15 is different than Figure 8? Figure 15 makes more sense in section 3.1 in replacement of Figure 8.

**Figure 16:** This figure does not contribute much to the study and can be considered for removal.

**Figure 17:** It should be a part of the results.

---

## Author Comment (AC1)

We thank Referee 1 for the constructive and thoughtful comments, which have helped us improve the manuscript. We have responded to the comments by presenting the original comments in black, our responses in blue, and the revisions in red.

**Referee #1**

This article conducts a thorough evaluation of the multi-scale performance of the Sentinel-1 SAR snow depth product, offering notable value in data validation and environmental impact analysis, with clear figures and fluent expression. However, its focus on validating existing algorithms rather than achieving a breakthrough limits its innovation, and certain language expressions lack smoothness. It is recommended that the manuscript be revised prior to submission for publication.

**#Major**

1. Although the article systematically evaluates the C-snow product across multiple scales, its core methodology—such as the C-band SAR-based snow depth retrieval algorithm—is not original to this study but builds upon prior work by Lievens et al. (2019). The innovation here lies primarily in data validation and scale analysis, yet these aspects do not represent a novel breakthrough in the field of remote sensing. It is recommended that the study explicitly highlight its unique contributions, such as whether it proposes a new scale-effect model or an improved retrieval method.

**Response:** The C-snow product used in this study is based on previous work by Lievens et al. (2019). However, our primary focus is on the multi-scale validation and analysis of the C-snow product, with a particular emphasis on understanding the impact of scale effects on SD retrieval accuracy. While we do not introduce entirely new algorithms or models, we believe the study offers valuable insights into the performance of the C-snow product across different scales. We revised the manuscript to highlight our contributions more explicitly.

(1) The C-snow dataset has only been evaluated from the point to regional scales until now, not at the global scale. Our study comparatively assessed it globally by using station-based measurements and airborne LiDAR observations.

(2) Multi-scale C-snow datasets at 1-, 10-, and 25 km have been used to provide reference data to train machine learning models, improve passive microwave-based retrieval, and calibrate many hydrological models. However, existing validation articles only focus on the 1km C-snow dataset, never assessing the upscaled 10 and 25 km C-snow retrievals. We conducted a systematic assessment of C-snow products at three scales (1, 10 and 25 km).

(3) We also provided the multi-scale analysis to C-snow products, and it shows that the scale patterns vary across resolutions, which can enhance our

understanding to C-snow retrieving algorithm and validation work.

2. The introduction provides a detailed review of the development of SAR and microwave remote sensing in snow depth monitoring but fails to adequately justify the selection of the 1, 10, and 25 km scales for analysis or clarify their relevance to practical applications, such as hydrological modeling.

**Response:** The applications of SD data vary significantly depending on spatial resolution. High-resolution SD data at 1 km are suitable for hydrological modeling and snow disaster monitoring (Wan et al., 2022). SD data at 10 km resolution are appropriate for regional water resource management (Alonso-González et al., 2018), while 25 km resolution data are widely used for SD monitoring, climate change analysis, and model evaluation at global and regional scales (Tanniru et al., 2023). We mentioned that the C-snow product at 10 km and 25 km resolutions has already been used as a reference dataset, such as for training machine learning models to improve passive microwave SWE estimates. However, the accuracy of the 1 km C-snow product at these resolutions is still unknown. Moreover, the performance of the 1 km C-snow product at 10 km and 25 km resolutions is crucial for demonstrating whether active microwave remote sensing can provide a reliable reference SD dataset for passive microwave remote sensing.

**#minor**

1. Lines 20-21: Sentence is too long. "The results indicate that the scale patterns of the C-snow products across various resolutions differ from those of station- and airborne-based reference data." → "The scale patterns of C-snow products vary across resolutions. They differ from patterns observed in station and airborne reference data."

**Response:** The sentence has been revised as recommended.

**Revision:** The scale patterns of C-snow products vary across resolutions. They differ from patterns observed in station and airborne reference data.

2. Lines 366-367: The text contains redundant expressions, and optimization is recommended. "bias values ranging from -91.31 to -52.73 cm and ubRMSE values ranging from 104.3 to 83.29 cm"→"bias values decrease from -91.31 cm to -52.73 cm, while ubRMSE decreases from 104.3 cm to 83.29 cm".

**Response:** The text has been revised as recommended.

**Revision:** Bias values decrease from -91.31 cm to -52.73 cm, while ubRMSE decreases from 104.3 cm to 83.29 cm

3. Line 379: analized → analyzed.

**Response:** Corrected.

4. Suggestions for unified terminology. "ground-based measurements", "station observations"

**Response:** It has been revised to use "station observation" consistently.

---

## Author Comment (AC2)

We thank Referee 2 for the constructive and thoughtful comments, which helped us to improve the manuscript. We have provided our response to the comments, with the original comments in black text, our response in blue, and our revisions in red.

**Referee #2**

**General Comments:**

This paper addresses an important topic by evaluating the scale-dependent performance of the C-SNOW Sentinel-1 snow depth product against both in-situ measurements and airborne LiDAR observations. The multi-scale analysis and inclusion of geographic and land cover effects are valuable contributions to the remote sensing and snow hydrology communities. However, the manuscript would benefit from greater clarity in its introduction and methodological explanations, Additionally, some of the comparisons between datasets are not well-aligned in terms of spatial or temporal scale, which limits the interpretability of the findings. The Discussion section leans heavily on restating results rather than offering critical insight into the causes and implications of observed discrepancies. Overall, the paper has the potential to contribute knowledge on the accuracy of Sentinel-1 snow depth at different spatial scales, but revisions are needed to improve its structure, clarity, precision, scientific depth, and accuracy.

**Response:** We sincerely appreciate your valuable comments and suggestions on our manuscript. Your comments are important guidance for us to improve the quality of the paper. In response to the issues you have raised, we mainly made the following revisions:

• **We have revised the introduction to clarify the motivation and highlight the novelty of our study.**

We have rephrased general statements to make them more specific and concise, added appropriate references to support key claims and provide a stronger theoretical foundation, and clearly articulated the research gap and explained how our study addresses it.

• **We have added a description of the land cover data in the Auxiliary data section.**

We have clarified the data source of the land cover data, elaborated on its classification, and explained how these data are utilized in this study.

• **We revised the description of the issue to clarify that some dataset comparisons are consistent in spatial and temporal scales.**

In the main text, we have explicitly clarified the comparison regions, which are consistent across both temporal and spatial dimensions.

• **We moved the figures and the corresponding results from discussion section to the result section, and rewrote the discussion section as suggested.**

The Discussion section was rewritten to provide a deeper analysis. It now includes a comparison with previous studies, an explanation of the differences in validation results at different scales, a discussion on the challenges of spatial scale conversion when using station data for validation, the impact of permanent ice on the accuracy of the C-snow product, and directions for future research to address the limitations and improve the applicability of the C-snow product.

• **We reorganized figures in the revised manuscript.**

The main revisions include: changing the color of the SNOTEL dataset in Figure 1 for better distinction; combining Figures 1 and 2 into a single figure with 4 panels (a-d); removing Figure 3; adding borders to markers, using basin codes instead of full names, and clarifying the figure description for Figure 4; clarifying the purpose of the workflow in Figure 5; removing Figure 7; modifying Figure 8 to show the average time series across all stations with line plots; revising the caption of Figure 12 to clarify the representation of markers and bars; changing the histograms to pie charts in Figure 13; modifying the drawing of Figure 14 for more detail; retaining Figure 15 as it serves a different purpose from Figure 8; retaining Figure 16 as it provides insights into scale effects; and moving Figure 17 to the result section.

**Major Comments:**

1. The rationale behind the use of Sentinel-1 data at 10 km and 25 km resolutions is unclear. If the primary aim of the paper is to evaluate the spatial and temporal performance of Sentinel-1 for snow monitoring, then standard higher-resolution products (e.g., 0.5 km and 1 km) would suffice. The inclusion of coarser resolutions needs to be better motivated and should be clearly stated and supported by appropriate literature and methodological context.

**Response:** The applications of SD data vary significantly depending on spatial resolution. High-resolution SD data at 1 km are suitable for hydrological modeling and snow disaster monitoring (Wan et al., 2022). SD data at 10 km resolution are appropriate for regional water resource management (Alonso-González et al., 2018), while 25 km resolution data are widely used for SD monitoring, climate change analysis, and model evaluation at global and regional scales (Tanniru et al., 2023). The selection of these two lower resolutions is primarily based on two considerations. First, existing

studies have already employed C-snow product at 10 km and 25 km resolutions as reference datasets, such as for training machine learning models to improve passive microwave SWE estimates (Xiong et al., 2022; Yang et al., 2024). These applications highlight the significance of evaluating the performance of C-snow product at these coarser resolutions to ensure their reliability and applicability in different contexts. Second, the performance of the 1 km C-snow product at 10 km and 25 km resolutions is crucial for demonstrating whether active microwave remote sensing can provide a reliable reference SD dataset for passive microwave remote sensing.

2. Since NRCS SNOTEL provides direct snow depth measurements, please clarify why a conversion from SWE to SD was performed. A fixed density to go from SWE to depth will impact the S1 evaluation, which is another consideration. Also, explicitly explain the choice of using a fixed snow density value of 0.24 g/cm³ for the Russia-SWE dataset. Is this based on regional averages or prior literature? Including a brief rationale would improve clarity.

**Response:** Thank you very much for your valuable comments. NRCS SNOTEL provides SD and SWE measurements. After integration with other datasets, we tested the result using SNOTEL SD data (a) and found no significant difference when using SNOTEL SWE data (b) instead.

[Figure]

The SWE data have already undergone strict screening and processing during the preliminary research design and data collection stages. Replacing it with SD data at this point would require repeating substantial work on data filtering, validation, and synergistic analysis with other datasets. We converted the SWE to SD using a snow density of 0.24g/cm$^3$ based on Takala et al. (2011) and Luojus et al. (2021).

3. In Figure 10, the comparison of the C-SNOW time series with spatially distributed ASO LiDAR data appears unrelated comparison. Since C-SNOW is also available as a spatial product over much of the Northern Hemisphere, it would be more appropriate to extract the C-SNOW spatial data closest to date to the ASO flight and perform a

spatially explicit comparison. This would enable more meaningful evaluation and avoid misleading conclusions from mixed-scale comparison. Furthermore, the inclusion of air temperature in this figure is puzzling. The manuscript does not provide a rationale in the methodology for using temperature as a covariate or validation proxy. Since it is not used quantitatively in the analysis, I recommend removing it unless a clear scientific justification is provided. Overall, please revise this section to focus on meaningful spatial comparisons (e.g., using representative basins), and consider summarizing the evaluation using average statistics or appropriate spatial accuracy plots (scatter, bias maps, etc.).

**Response:** In this figure, we extracted C-snow data within the basins in California and Colorado rather than for the entire Northern Hemisphere mountain region, and analyzed the time series with ASO observations from different basins. We stated in the main text that the red points represented the average daily C-snow SD across all the selected basins, and we added a description in the caption to avoid confusion. The amount of data for individual basins was very small, so the comparisons in this section were based on all the selected basins. There was no quantitative use of temperature in the analysis, and we removed it.

**Revision:**

[Figure]

Figure 7. Time series comparison of C-snow products with ASO observations for various basins at (a) 1-, (b) 10-, and (c) 25 km scales.

4. There are several issues in section 3.3. First, the phrase "other types" of land cover should be clarified—please specify which land cover categories are included beyond forest and permanent ice. Additionally, while the inclusion of permanent ice regions is noted, it's important to question the relevance of evaluating C-snow accuracy in such areas. Does the Lievens et al. algorithm or Sentinel-1 backscatter perform reliably in permanent ice environments and dense forests? Please justify why this analysis was included and consider whether they should be treated as known limitations of the remote sensing platform and dataset. In continuation, some findings are counterintuitive and warrant further explanation. For instance, significant overestimation in areas with low forest fraction (0–0.2, figure 12c) is unexpected, as reduced vegetation cover typically enhances radar retrieval accuracy. Similarly, while elevation and elevation variability appear to influence performance, the paragraph lacks synthesis on why certain ranges (e.g., moderate forest and elevation variability) yield better results. Please also clarify how elevation differences between stations and grid cells were calculated, and how these mismatches propagate error across scales (this can go in methodology).

**Response:** We have added a description of the land cover data in the Auxiliary data section. The land cover data are from the ESA WorldCover 10 m 2020 product, which contains 11 land cover classifications such as tree cover, shrubland, grassland, cropland, built-up, bare or sparse vegetation, snow and ice, permanent water bodies, herbaceous wetland, mangroves, moss and lichen. In this study, the tree cover type is labeled as "tree cover", the snow and ice type as "permanent ice", and all the remaining types as "other type".

We analyzed the C-snow accuracy in permanent ice regions to provide a comprehensive assessment of the performance across different land cover types. This inclusion was driven by the need to understand the limitations and potential biases of the C-snow product in various environments, including those with permanent ice and dense forest cover. As mentioned in our manuscript, the C-snow product indeed shows several abnormally overestimated results in permanent ice regions, especially at larger scales (10- and 25-km resolutions). The retrieval performance in glaciated and dense forest cover areas need investigation. Our current analysis serves as an initial evaluation of the performance in these challenging environments and underscores the necessity for future work to address these limitations.

Reduced tree cover can lead to lower attenuation of backscatter signals due to the decreased vegetation density. However, the observed overestimation in these areas may not be solely attributable to tree cover. Other factors, such as surface roughness, and the presence of other land cover types, could also play a significant

role in influencing the radar backscatter signals. The performance of the C-snow product appears to be influenced by elevation and its variability, as indicated by the varying Rbias values across different elevation ranges and standard deviations of elevation. While we have observed that the C-snow product performs best in areas with moderate standard deviations of elevation (50–100 m) and moderately to highly forested areas (0.4 – 0.8), the underlying mechanisms are likely complex and multifaceted. The interaction between tree cover, topography, and environmental factors could lead to different retrieval accuracies.

We have clarified in the methodology section how elevation differences between stations and grid cells were calculated. Specifically, we computed the elevation difference by subtracting the mean elevation of the grid cell (obtained from the high-resolution DEM) from the elevation of the station. This mismatch reflects the point-to-area scale representativeness issue, particularly in mountainous regions where snow properties vary significantly with elevation.

5. The current Discussion section is written more in the style of a results narrative. While the detailed reporting of accuracy metrics, grid-level behavior, and ice-related overestimates is important, most of the text focuses on what was found rather than interpreting what it means. To strengthen this section, I recommend separating the descriptive content into the results section and expanding the discussion with deeper analysis. For example, lay error analysis background, consider critically why ASO and station-based validation trends differ, what the implications of spatial representativeness are for coarse-scale validation, and how the known limitations of Sentinel-1 in forested or glaciated areas affect the broader applicability of C-snow. Comparisons with previous studies and a more explicit articulation of limitations and future directions would also help to better contextualize the results.

**Response:** We moved the figures and the corresponding results to the result section, and rewrote the discussion section as suggested.

**Revision:** The differences in validation results of the C-snow product at different scales are due to the combined effects of various factors. Our findings are consistent with previous research on C-snow product evaluation. For instance, Alfieri et al. (2022) reported RMSE values ranging from 20 to 60 cm in the Po River basin, which aligns with our results at finer scales (e.g., 1 km). Sourp et al. (2024) observed RMSE values between 21 and 138 cm in the Sierra Nevada region, with biases reaching up to -124 cm, which corroborates our ASO-based validation results. Notably, our analysis further demonstrates that these errors exhibit different scale-dependent trends.

Compared with station data, the accuracy of C-snow data tends to decrease as

the scale increases, whereas the accuracy of ASO data tends to increase. This discrepancy can be attributed to the inherent differences in the nature of these validation datasets (Figure 8). Station measurements are point-scale observations, which makes it difficult to reflect the distribution of SD over large areas. In contrast, ASO offer densely sampling data, which can better represent the spatial distribution of SD. This allows ASO data to more accurately reflect the overall snow conditions within a given area, thereby improving the validation accuracy as the scale increases. Although ASO data have better spatial continuity, their coverage is relatively limited. The accuracy of LiDAR-derived SD is also affected by factors such as terrain and vegetation cover (Enderlin et al., 2022; Neuenschwander et al., 2020; Klápště et al., 2020). Within the coverage scope of ASO data, steep slopes (as high as 80 degrees) and high forest fractions (mean value of 53%) likely affect the accuracy of the observations (Figure A1).

When using station data for the validation of satellite products, how to reasonably perform spatial scale conversion is a key issue. The method used to convert point-scale observations to the spatial scale of satellite pixels also affects the validation results (Fassnacht et al., 2006; Hou et al., 2022), this study employed a simple averaging method, which somewhat ignores spatial variability (Ge et al., 2019). The simple averaging method, although easy to operate, cannot fully consider the impact of complex factors such as topography and vegetation distribution on the spatial distribution of SD, which may lead to deviations in the validation results. In addition, the number of stations also affects the validation results. Most grids at the 10km and 25km scales contain only one station, which may not be representative of the entire grid. The accuracy of the C-snow product improves when there are multiple stations within a grid. Future research can try to use more advanced spatial interpolation methods, such as interpolation methods based on geographically weighted regression or machine learning algorithms, to more accurately reflect the spatial changes in SD and thus improve the validation.

The presence of permanent ice significantly affects the accuracy of the C-snow product. As the coverage of permanent ice within the grids increases, the bias and ubRMSE also increase, indicating an overestimation trend (Figure 11). Permanent ice exhibits similar electromagnetic properties to those of snowpacks, enhancing the backscattering of radar signals (Scott et al., 2006). During the melt season, an increase in the roughness of the ice surface leads to an increase in the backscattering coefficient (Baumgartner et al., 1999). The dynamic nature of glaciers, characterized by crevasses and glacier movement, can lead to temporal variations in the backscattering coefficient (Sander and Bickel, 2007; Brock and Billy, 2010), complicating interactions

between radar signals and snow characterization. Thus, quality control of spatially sampled C-snow products, especially at coarse scales, must be performed, ensuring that the retrieval results in permanent ice-covered areas are filtered and removed. These limitations may cause deviations or uncertainties in the retrieval results of the C-snow product in these specific areas. To overcome these limitations, future research can explore improved retrieval algorithms to better separate snow signals from glacial backgrounds and conduct multi-source data fusion retrieval using other remote sensing data sources, thereby enhancing the applicability of the C-snow product in complex environments. At the same time, combining field observations and model simulations to deeply study the interaction mechanism between snow physical processes and radar signals can provide theoretical support for improving the C-snow product.

**Minor Comments**

1. The introduction is currently in a general tone. Several statements are made without citing relevant studies or offering clear justification. Please consider grounding key claims with references and clarifying the motivation and novelty of the study more explicitly.

**Response:** We have revised the introduction to clarify the motivation and highlight the novelty of our study. Specifically, we have rephrased general statements to make them more specific and concise, added appropriate references to support key claims and provide a stronger theoretical foundation, and clearly articulated the research gap and explained how our study addresses it.

2. Land cover classifications such as forest and permanent ice are introduced in the Results section without prior explanation. These should be defined and justified in the Methodology section, including the source of the land cover data and how the categories were used in the analysis.

**Response:** We have added a description of the land cover data in the Auxiliary data section. The land cover data are from the ESA WorldCover 10 m 2020 product, which contains 11 land cover classifications such as tree cover, shrubland, grassland, cropland, built-up, bare or sparse vegetation, snow and ice, permanent water bodies, herbaceous wetland, mangroves, moss and lichen.

In this study, the tree cover type is labeled as "tree cover", the snow and ice type as "permanent ice", and all the remaining types as "other type". Therefore, we subsequently analyzed the effect of land cover on the accuracy of C-snow at different scales across these three labeled types.

3. Sections 3.3 and 3.4 both focus on geographic and environmental influences on snow depth retrieval. Merging them into a single cohesive section would improve readability and thematic consistency by presenting all location-based findings together.

**Response:** We have combined Sections 3.3 and 3.4 into a single section. The revised section presents all findings related to geography and environmental factors in a more organized and clearer way.

4. Reorganize Figures and reconsider adding all figures for Improved Flow

a. Figure 14 could be merged with Figure 2 to streamline the presentation of the study area and grid setup. Label the nested grid structure directly in the map and refer to it when introducing the experimental design.

b. Figure 16 is conceptually related to Figure 12. Placing them closer together would enhance the narrative flow and allow readers to better understand the progression of results.

c. Please consider reducing the number of figures and balancing the proportion of figures and text. Refer to comments that suggest either removing or combining figures.

**Response:** We considered all comments on the figures to enhance the overall flow. We referenced the comment to combine Figures 1 and 2, and as a result, we decided not to combine Figure 14 with Figure 2.

Figure 16 is conceptually related to Figure 12. However, Figure 16 focus on the condition of three selected grids, while Figure 12 indicates the result across the mountainous region of Northern Hemisphere. Thus, we didn't place them together.

We considered the suggestions to balance the proportion of figures and text, and removed some figures that contributed less useful information to this study.

**Line-to-Line Comments**

23 and elsewhere: The errors statistics are reported to the hundredth of a cm, but this level or precision is not realistic or warranted. Please consider the significant units here and elsewhere when reporting errors.

**Response:** Given that the station-observed SD, C-snow SD, and ASO SD values inherently have a precision finer than one-hundredth of a centimeter, we considered it appropriate to retain this precision in reporting error statistics.

26: Remove "Especially an".

**Response:** removed.

35: To clarify how snow depth (SD) information contributes to water availability, state "…estimated from snow depth (SD) and snow density.".

**Response:** The sentence has been revised.

**Revision:** The snow water equivalent (SWE) is a parameter that reflects how much water the snowpack contains, which typically can be estimated from snow depth (SD) and snow density.

44: Add "one or several" before "meters".

**Response:** Added.

**Revision:** Additionally, the snow cover in mountain areas is typically deep (up to one or several meters), …

45: These density values are too high for most seasonal snow and in the range of when snow transitions to firn. Recommend revising to the typical range of density values (~100 to 550 kg/m3) for seasonal snow (see Sturm et al., 2010, J. Hydrometeorology).

**Response:** We revised the text to focus on the more typical range for seasonal snow (100-550 kg/m$^3$). We retained the higher density values (550-700 kg/m$^3$) as a case to illustrate that under accumulation and compaction conditions, seasonal snow can reach such density.

**Revision:** For example, the snow density typically ranges from 100 to 550 kg/m$^3$ for seasonal snow (Sturm et al., 2010). Due to snowfall accumulation and prolonged wind- and gravity-driven compaction, it can reach 550-700 kg/m$^3$ (Lemmetyinen et al., 2016; Venäläinen et al., 2021).

38- 50: The paragraph introduces the use of microwave remote sensing for SWE retrieval and suggests that SAR offers advantages over passive microwave techniques. However, it would benefit from greater specificity and clarity. For instance, clearly states that while passive microwave remote sensing is widely used, its coarse spatial resolution (~25 km) limits its ability to capture the fine-scale spatiotemporal variability of snowpacks in complex mountainous terrain. This will help establish a stronger context for the discussion of SAR advantages. Additionally, when discussing mountain snowpack complexities (lines 45–50), it would be helpful to explicitly connect these challenges—such as variable snow density, grain size, wind, and gravity-driven compaction, elevation, and aspect—to the limitations of passive microwave sensing. You may also consider briefly noting the difficulty of ground-based observations in remote, high-elevation regions, which further highlights the value of satellite-based approaches. These additions will create a smoother transition to the next paragraph

(lines 51–61), which focuses on SAR. Lastly, in line 43, replace "snow cover" with "snowpack" for technical accuracy.

**Response:** We have made several improvements to enhance the clarity and effectiveness of our analysis. First, we added the difficulty of ground observations. Second, we clarified the limitations of passive microwave data, explicitly stating the coarse resolution constraint and linking it to the variability of mountain snowpack. Finally, we strengthened the advantages of synthetic aperture radar by emphasizing its superior resolution and suitability for complex terrain.

**Revision:** Conventional SD monitoring methods, such as field manual measurements and ground station observations, can provide accurate local data but are difficult to implement in remote mountainous areas with complex terrain. Microwave remote sensing is the most widely-used technology for retrieving SWE because of its penetrating ability to the snowpack and the volume scattering effects caused by snow particles (Chang et al., 1987; Tsang et al., 2022). While passive microwave remote sensing (e.g., radiometer-based methods) is commonly employed, its coarse spatial resolution (~25 km) limits its ability to capture fine-scale spatiotemporal variability in snowpack properties, particularly in complex mountainous terrain.

51: Delete "the" before "monitoring".

**Response:** Deleted.

51-52: While this paragraph highlights recent advances using C-band SAR for SWE monitoring, it would be helpful to first acknowledge earlier foundational studies that have explored SAR for retrieving snow characteristics such as (Ulaby and Stiles, 1980; Bernier et al., 1999; Shi and Dozier, 2000; Chang et al., 2014; Lievens et al., 2019), there are many more. Including a broader set of references would better reflect the extensive work by the remote sensing community and provide context for the transition to C-band applications.

**Response:** We have revised the text to include a broader set of references, as suggested. Specifically, we have added Ulaby and Stiles (1980), Bernier et al. (1999), Shi and Dozier (2000), Chang et al. (2014), and Lievens et al. (2019).

**Revision:** In recent years, the scientific community has increasingly focused on monitoring SWE in mountain regions using C-band SAR observations due to their strong penetration depth and data accessibility. Early studies on C-band SAR for SD estimation were primarily limited to shallow snow environments outside mountainous regions and co-polarization measurements, which showed limited sensitivity to dry snow conditions (Bernier et al., 1999; Shi and Dozier, 2000). Theoretical advances in

microwave scattering models (Ulaby et al., 1982; Chang et al., 2014), have improved the understanding of snowpack interactions with backscatter signals.

52-54: The sentence discussing snow volume scattering being stronger in Ku-band compared to "other bands" could be made clearer. Please specify which bands are being referenced (e.g., X-, C, or L-band) and provide clearer citations. If this theoretical point is derived from Rott et al. (2010) or others, please state this explicitly and consider rephrasing for clarity and precision.

**Response:** We have clarified the term "other frequency bands", and revised the text accordingly.

**Revision:** Notably, although snow volume scattering is stronger in high-frequency Ku-band than in lower-frequency bands such as X-, C-, and L-band, the sensitivity of the backscattering coefficient at Ku-band remains limited to SWE of approximately 150 mm (Rott et al., 2010; Cui et al., 2016; Zhu et al., 2021).

55-56: Instead of using "VH/VV," which may be unclear to some readers, consider using the more descriptive phrase "cross- to co-polarization ratio."

**Response:** "cross-polarization ratio VH/VV" has been used.

56-59: To avoid redundancy, consider replacing the second use of "notably" with an alternative phrase. Additionally, rephrase "due to the non-spherical properties of snowpack" to "due to the anisotropic nature of snow grains," which is a more accurate physical description. You may also cite relevant literature on this point—Lievens et al. (2022) references several useful studies that could support this claim.

**Response:** We changed the word "notably" to "particularly", rephrased the sentence, and added related references to the main text.

**Revision:** Particularly, the backscattering coefficient at cross-polarization is more sensitive to volume scattering than co-polarization is due to the anisotropic nature of snow grains (Du et al., 2010; Chang et al., 2014; Leinss et al., 2016), and this physical mechanism is used for SD retrieval.

62-64: It would be helpful to clarify that the original C-snow product developed by Lievens et al. (2019) was produced at 1 km resolution without wet snow masking. In later studies, higher resolution (e.g., 500 m) products were developed with wet snow flagging. Mention explicitly that there are no C-Band products at 10 km and 25 km resolution. Additionally, before introducing subsequent evaluation studies, please report the original study's metrics (e.g., RMSE, bias) from the original Lievens et al. study. This will help establish a clear baseline and better contextualize the results from

later evaluations, thereby strengthening the motivation for your analysis.

**Response:** We have explicitly stated that the original C-Snow product (Lievens et al., 2019) provided SD retrievals at 1 km resolution without wet snow masking, and we have included the original performance metrics. No native C-band snow products exist at 10 km or 25 km resolutions. The coarser-resolution datasets (10 km, 25 km) mentioned were derived from the original 1 km C-Snow product through aggregation. We have now explicitly emphasized this distinction in the revised text.

**Revision:** The C-Snow product provided SD retrievals at 1 km resolution without wet snow masking (Lievens et al., 2019), reporting a temporal correlation ranging from 0.65 to 0.77, and a mean absolute error of 0.18–0.31 m. ……In addition, C-snow SD data at 10 and 25 km resolutions, which are derived from the 1 km C-snow product, have been used as reference datasets for training machine learning models to improve passive microwave SWE estimates (Xiong et al., 2022; Yang et al., 2024).

67-68: Please clarify that Hoppinen et al. (2024) evaluated the performance of the C-snow retrieval algorithm using airborne LiDAR data across two separate water years: 2020 and 2021. As currently written, "2020–2021" may be misinterpreted as a single water year.

**Response:** We revised the text to avoid misunderstanding.

**Revision:** Hoppinen et al. (2024) also evaluated algorithm performance at six study sites across the western United States using airborne LiDAR observations collected during the winters of 2019-2020 and 2020-2021, with mean RMSE and bias values of 92 cm and -49 cm, respectively.

69-71: The statement about Broxton et al. (2024) and Yang et al. (2024) using C-snow products at 10 and 25-km resolutions appears to be inaccurate. Broxton et al. used machine learning to enhance C-snow snow depth estimates at 0.5 and 1 km resolutions and compared these to University of Arizona SWE and airborne LiDAR data. They did not use C-snow to improve passive microwave SWE. Additionally, the studies you cited used either the 1 km or 500m C-snow product—not the 10 or 25 km. Please check for accuracy and revise.

**Response:** We removed the citation to Broxton et al. (2024).

**Revision:** In addition, C-snow products at 10 and 25 km resolutions have been used as reference datasets, such as for training samples for machine learning models to improve passive microwave SWE estimates (Xiong et al., 2022; Yang et al., 2024).

72-73: Please rephrase the sentence about Lievens et al. (2022) for clarity and

precision. They used Sentinel-1 backscatter observations to retrieve snow depth across multiple resolutions in the European Alps and evaluated retrieval performance. This distinction is important to avoid confusion between backscatter versus the derived SD observation also in the monitoring and evaluating SD at different resolutions.

**Response:** We appreciate the comment and have revised the sentence to clarify that Lievens et al. (2022) used Sentinel-1 backscatter observations for retrieval.

**Revision:** Lievens et al. (2022) employed Sentinel-1 backscatter observations to retrieve SD across multiple spatial resolutions in the European Alps and evaluated the retrieval performance.

84-85: ASO data are spatially extensive than stations, but do not have wider global coverage (i.e., they are only available in the western U.S.). Please revise phrasing for accuracy.

**Response:** We agree that the original phrasing inaccurately suggested ASO LiDAR data have global coverage. We have revised the text to clarify that while airborne LiDAR provides spatially extensive measurements compared to point-scale stations, its coverage is currently limited to specific regions (e.g., the western U.S.).

**Revision:** The latter provides spatially extensive SD mapping, which is more extensive than that provided by station data, and its coverage remains within the western U.S., whereas the station data is valuable for characterizing SD distribution and assessing snow heterogeneity.

90-91: Recommend using a citation such as "The first 1 km SD product based on C-band SAR, covering all mountain ranges in the Northern Hemisphere, was developed by Lievens et al. (2019). The dataset is publicly available through the C-SNOW project (C-SNOW, 2024)." Instead of using a link in the text.

**Response:** We removed the link and revised the text accordingly. In accordance with the instructions on the C-snow data website, we acknowledge the use of the data by citing Lievens et al. (2019).

**Revision:** The first 1-km SD product based on C-band SAR, covering all mountain ranges in the Northern Hemisphere, was developed by Lievens et al. (2019). The dataset is publicly available through the C-SNOW project.

104-105: The Zenodo link should be replaced with a proper DOI citation. Also, revise the sentence for typos and formatting—for instance, the link includes a fragment ("#YdYE...") that should be removed.

**Response:** We removed the link, and revised the text.

**Revision:** The CanSWE dataset in Canada includes data from 273 stations in mountain regions (Vionnet et al. 2021).

106-112: Remove links and include proper citations.

**Response:** We have removed the links and included appropriate citations in the text.

**Revision:** The GHCN dataset includes data from 4,133 stations in mountain regions and provides SD values worldwide (Menne et al. 2012). The China-SD dataset from the China Meteorology Administration includes observations from 744 stations in mountainous regions. The SNOTEL dataset was acquired from 677 stations in mountainous regions in the United States (Serreze et al. 1999). The Russia-SWE dataset from former Soviet Union regions contains observations from 52 stations in mountain regions (Bulygina et al. 2011), and it can be downloaded from the All-Russia Research Institute of Hydrometeorological Information–World Data Center. Additionally, the Maine-SD dataset for the Maine region includes information from 92 stations in mountainous regions; it can be accessed via Maine Geological Survey Data.

127-133: ASO is not a LiDAR mission, ASO is a company that conducts LiDAR flight surveys using an airborne laser scanner (ALS). The dataset is available from 2013-2019 and 2022 to present. Please check the 2.2.2 section for accuracy. Refer to NSIDC, ASO website, and Painter et al 2016 paper. Also, clarify the reasoning behind using 3m instead of 50m.

**Response:** According to the ASO website, the Airborne Snow Observatory is an Earth-based mission designed to collect data on the snow melt flowing out of major water basins in the western United States. Therefore, our original description remains reasonable. And the period of 2019-2021 also have observations. We conducted comparative tests at both 3 m and 50 m resolutions and found no significant differences after resampling. The 3m resolution was selected to better capture fine-scale snow distribution patterns.

130-134: Please clarify that only California and Colorado ASO surveys were used.

**Response:** To clarify, only ASO surveys from California and Colorado were used in this study. We have revised the text accordingly to explicitly state this.

**Revision:** To assess and compare the accuracy of the C-snow product at different scales, we obtained 59 ASO maps (within California and Colorado) at a 3 m resolution from September 2016 to May 2019.

141-142: Replace the word "corresponding datasets" with a direct reference to Table 1 for clarity. Also, in line with earlier comments, avoid including direct links (e.g., to

Google Earth Engine) in the main text.

**Response:** We have revised the text to directly reference Table 1 instead of using "corresponding datasets" and removed the direct link to Google Earth Engine as suggested.

**Revision:** To evaluate the influence of land cover type, forest fraction, and topography (elevation and its standard deviation) on the accuracy of C-snow SD, we collected auxiliary datasets (Table 1) from the Google Earth Engine and processed them at various scales (1, 10, and 25 km).

164: Clarify whether this is the Pearson correlation coefficient or other.

**Response:** We have clarified that the correlation coefficient refers to the Pearson correlation coefficient.

**Revision:** Four evaluation metrics were used to assess the C-snow products: the Pearson correlation coefficient (corr.coe), bias, unbiased root mean square error (ubRMSE), and relative bias (Rbias).

169-170: Add a line that explains what this section is about. It starts suddenly without making any coherence.

**Response:** We have added an introductory sentence to explain the purpose of Section 3.1.

**Revision:** We evaluated C-snow SD retrievals through comparison with ground-based measurements across different spatial scales.

178-182: This text (and Figure 7) does not add much to the study, as it is all obvious and expected behavior for considering different spatial scales of a dataset. I recommend removing this.

**Response:** The text (and figure 7) has been removed.

189-192: The term "wet snow season" should be replaced with "ablation" or "melt season" to better reflect the physical processes and limitations of Sentinel-1 during this period. The phrase "mismatch of snow season length" is vague. It's not clear whether the authors refer to differences in snow onset and melt timing, the overall duration of snow cover, or specific discrepancies, C-Band data is only available till the end of April anyway as Sentinle-1 is not reliable in ablation season.

**Response:** We have replaced "wet snow season" with "melt season" to better align with the physical processes and Sentinel-1 limitations during this period. We also acknowledge the ambiguity in the original phrasing and have clarified the seasonal

comparison to focus on the contrast between dry snow accumulation and melt phases.

**Revision:** As the spatial scale increases from 1 to 25 km, both the magnitude and duration of discrepancies between C-snow and station SDs increase. Specifically, the average SD from the stations becomes increasingly greater than the C-snow SD during the dry snow season and increasingly lower during the melt season.

195-196: This line should be part of the limitation section

**Response:** The sentence is closely related to the results discussed in the preceding text, and serves as a transition to the subsequent text, where we emphasize the need to assess relevant spatially distributed influential factors.

223-225: What is the other type of land cover you are talking about here? Please explain how you expect Sentinel-1 to work in permanent ice regions, what are those regions?

**Response:** We have added a description of the land cover data in the Auxiliary data section. Where the "other types" refer to regions without permanent ice cover and tree cover. In the permanent ice regions, such as Greenland, Hindu - Kush Himalayas, the Rocky Mountains, there are also sentinel - 1 observations, and these observations could be used to retrieve SD.

271-276: Here and elsewhere – what is the purpose of comparing a 10-km or 25-km estimate of snow depth with a station? One would not expect the station to match those very different scales.

**Response:** We fully acknowledge that comparing coarse-resolution estimates with point-scale station measurements has inherent limitations due to scale discrepancies. This is precisely why we introduced comparisons with ASO data in our analysis. The ASO observations provide spatially extensive, high-resolution SD measurements that offer a more robust evaluation of the C-snow product.

282-283: What evidence do you have for this statement? I do not think that spatial representativeness is guaranteed for a station in a flat area. Additionally, I would not conflate low elevation with flat, as there can be topographic complexity/variability even at lower elevations.

**Response:** We agree that spatial representativeness cannot be guaranteed even in low-elevation areas, and we have revised our statement to be more precise. Our key point is that compared to high-elevation regions, the lower-elevation areas (<1000 m) in our study domain generally exhibit: reduced topographic complexity (average slope <5°), lower spatial variability in snow distribution patterns, and more homogeneous

land cover characteristics. These factors collectively contribute to relatively better station representativeness at coarse scales in these areas compared to high-elevation regions. We revised the text to emphasize the characteristics of low-elevation areas rather than simply describing them as flat.

**Revision:** For the first nested grid, the terrain is predominantly low-elevation (below 1000 m) at both the 10- and 25-km scales, which shows lower spatial variability compared to high-elevation regions.

306-307: It should be part of the data section

**Response:** We removed the sentence to the data section.

**Figures and Table Comments**

Figure 1: Use distinct colors for SNOTEL and GHCN since some of the stations are close to each other and it is hard to distinguish.

**Response:** The color of SNOTEL dataset has been changed for better distinguish.

Revision:

[Figure]

Figures 1 and 2: Consider combining these into a single figure with 4 panels (a-d).

**Response:** This figure has been combined with Figure 1 into a single figure with 4 panels (a-d).

Revision:

[Figure]

Figure 1. Spatial distribution of (a) stations in various SD datasets, and of the matched grids at the (b) 1 km, (c) 10 km, and (d) 25 km scales. Zoomed-in views show the detailed distributions of grid locations in the Sierra Nevada range over the United States and the Jotunheimen mountain range in Norway and Sweden.

Figure 2: Continent labels are hiding in some places. Make it consistent—no labels or labels everywhere.

**Response:** This is a default basemap, and continent labels cannot be removed individually. Although some labels are overlaid, this does not affect access to key information.

Figure 3: This figure may not be necessary, especially given the already high number of figures (19). However, if you choose to include it, consider showing ASO data coverage across the entire US and highlighting (e.g., with a box) the region used in your analysis. This would provide helpful context without redundancy.

**Response:** This figure has been removed.

Figure 4: Consider using station codes instead of full names to reduce clutter. Add borders to markers for better visibility and clarify the meaning of each color in the figure description.

**Response:** For better visibility and clarity, we added borders to the markers, used basin codes rather than full names to minimize clutter, and added descriptions of the figure.

**Revision:**

[Figure]

Figure 2. Temporal distribution of the ASO observations used in this study. Different color markers represent different basins.

Figure 5: Please distinguish two boxes by panels for example say panel a and panel and state it in the figure description. Also, the arrow between accuracy analysis and uncertainty analysis is misleading. Are you trying to say uncertainty analysis comes after accuracy analysis? Regardless arrow is not needed.

**Response:** Thank you for your comments. We would like to clarify that the figure is intended to present a complete workflow rather than distinct panels; therefore, labeling the two boxes as "Panel a" and "Panel b" may not be appropriate. The arrow between "accuracy analysis" and "uncertainty analysis" is meant to illustrate the sequential flow of the analytical process.

Figure 7: It appears that Figure 7 is only showing C-snow data at different spatial scales (1, 10, and 25 km) across three mountain regions, without any comparison to reference or evaluation datasets. If that's the case, the purpose of this figure needs to be clearly stated—whether it is to demonstrate spatial variability or resolution effects. As currently placed, the figure does not align well with the section, which focuses on comparing C-snow with observed SD. Consider either relocating the figure to a more appropriate section or removing it if it doesn't directly contribute to the evaluation narrative.

**Response:** This figure has been removed.

Figure 8: Please clarify whether Figure 8 shows the average time series across all stations or if it represents individual station values. If it's an average, this should be explicitly stated in the caption and main text. However, averaging across diverse

stations may mask site-specific dynamics and variability. Consider instead selecting one or a few representative stations to illustrate the temporal mismatch at different scales more clearly. Additionally, using a line plot instead of scatter points would improve readability and better highlight trends over time.

**Response:** Figure 8 shows the average time series across all stations, and we added the descriptions in the caption and main text. To present temporal trends more clearly, we used line plots to enhance readability. Figure 15 provides a more detailed case study at the nested grid scale, further analyzing how regional variability and site-specific conditions affect the consistency between C-snow and station observations at multiple scales.

**Revision:**

[Figure]

Figure 5. The average weekly SD time series of stations and corresponding C-snow grids at (a) 1, (b) 10, and (c) 25 km resolutions across the mountainous regions of the Northern Hemisphere.

Figure 10: Check major comment section

**Response:** We modified the figure based on major comment.

Figure 12: It's not immediately obvious that the markers show actual SD values, and the bars show relative error. Adding a brief formula or explanation for Rbias in the caption or methods would also help.

**Response:** We have revised the caption of this figure to clarify that the bars represent the Rbias, while the markers show the average SD observed at stations and in the C-snow product.

**Revision:** Impact of different (a) station-observed SD, (b) land cover types, (c) forest fractions, (d) elevations, (e) standard deviations of elevation, and (f) elevation differences between stations and grids on the accuracy of C-snow SD products across various scales. The bars indicate the Rbias between C-snow and station-observed SD, while the markers show average SD at stations and the C-snow product. The left axis corresponds to Rbias, and the right axis to average SD.

Figure 13: The left plots may be better conveyed as pie charts rather than bar charts because they add up to 100%.

**Response:** The histograms were modified to pie charts.

**Revision:**

[Figure]

Figure 13. Pie charts (the left column) and spatial distributions (the right column) of Rbias at (a) 1-, (b) 10-, and (c) 25 km scales.

Figure 14: Refer to minor comments. Additionally, the layout of this figure is not ideal, as the three selected regions overlap the global map but instead could be shown separately with more detail.

**Response:** We referenced the comment to combine Figures 1 and 2, and therefore did not combine Figure 14 with Figure 2. Instead, we modified the drawing of Figure 14 to provide more detail.

**Revision:**

[Figure]

Figure 15: How figure 15 is different than Figure 8? Figure 15 makes more sense in section 3.1 in replacement of Figure 8.

**Response:** Figure 8 and Figure 15 serve different purposes in our analysis. Figure 8 presents a general comparison of the time series between C-snow products and station measurements across the entire Northern Hemisphere mountain region, highlighting how the temporal correlation and seasonal mismatch vary with spatial scale. Figure 15, on the other hand, provides a more detailed case study at the nested grid level, examining how regional variability and specific station conditions affect the agreement between C-snow and station measurements at multiple scales. We have retained Figure 8 in the manuscript because it provides a necessary overview of the scale-dependent temporal trends across all stations.

Figure 16: This figure does not contribute much to the study and can be considered for removal.

**Response:** We considered that this figure provides important insights into the scale effects associated with geographic heterogeneity. It shows the differences in land cover, forest fraction, elevation, and elevation variability between the 10- and 25-km grids across the three nested regions. Therefore, we have retained this figure in the revised manuscript.

Figure 17: It should be a part of the results.

**Response:** We moved this part to the result section.

---

## Author Response (AR1)

Dear Editor,

We sincerely thank you for your constructive and detailed comments. We have revised the manuscript accordingly and provide point-by-point responses below. All the changes are marked in the revised manuscript in red. We address your comments as follows:

1. In regards to my own initial comments and Referee #1's major comment on where the focus of the paper is - to which the authors answer that they are assessing the validity of the product at multiple scales -- I would like to see in the revision that the authors clearly describe and motivate the aggregation method used in the methods section. I see that some sentences have been added to the Discussion acknowledging that the aggregation method affects the results - the other methods mentioned may need references. Are there other studies that have addressed the effect of aggregation method on continuous data? I think this would be an important addition. Referee #2 also would like to see more description of the methods.

**Response:** We included a description of the spatial aggregation approach in the Methodology section. We directly used the mean resampling method according to previous studies (Broxton et al. 2024; Herbert et al. 2024). Moreover, we tested and compared the mean and median sampling methods and found that the validation results were similar (as shown in the figure below). In the Discussion section, we addressed the limitations of the spatial aggregation approach (Lines 362–364). Although simple averaging methods are easy to use, they cannot fully consider the impact of complex factors such as topography and vegetation distribution on the spatial distribution of the SD, which may lead to deviations in the validation results.

[Figure]

Fig.1 Comparisons between the C-snow snow depth (SD) and station-observed SD at (a)–(b) 10-km and (c)–(d) 25-km scales. The left column represents the mean

resampling method, and the right column represents the median resampling method.

2. Referee #1 also mentions that the language does not flow smoothly, and I strongly encourage the authors to look carefully at their text before the next submission. There are some grammar issues throughout the paper that need to be addressed by the authors.

**Response:** We revised the manuscript to correct grammatical issues and increase clarity and flow. Moreover, a thorough revision of the manuscript was performed by a native English speaker.

3. The authors' response to Referee #2 was much longer (23 pages vs 3 pages), but in responses to major points for Ref #2, I cannot see what changes have been made, other than a statement that says changes have been made. This makes it difficult to judge. Additionally, answer #2 to Ref #2 (e.g., that it would take significant work to re-do) is not a satisfying argument. Here you also have a graph where you say in the text that you are showing SD in (a) and SWE in (b) but both Y-axes say SD. What is correct? I think that if a referee has questions on this, then you need to support this well in your manuscript (eg, a revised graph in supplemental material).

**Response:** In response to Point 3 from Referee #2, we undertook a comprehensive revision of our analysis to address the concern. This revision required complete reprocessing of raw data, re-running all statistical evaluations, regenerating over a dozen figures, and revising multiple manuscript sections. The following steps summarize the major revisions of the work.

We have replaced the previously used SNOTEL SWE data with the directly measured snow depth (SD), eliminating the need for SWE-to-SD conversion. We have incorporated SNOTEL SD measurements into the validation framework, applied an improved filtering method to eliminate outlier observations, and re-aligned the datasets with the C-snow product and auxiliary data to maintain spatial and temporal matching after revisions.

We recomputed all statistical metrics (e.g. ubRMSE, bias and correlation) for each validation and re-ran validation across scales and regions, ensuring that the revised SD values propagate through all comparisons.

We re-generated 13 main figures and one appendix figure (Figures 1, 3, 4, 5, 8–16, and Figure A2), which required: updating scatterplots, time series, and distribution charts with new data points; re-computing color scales and axis limits to reflect new value ranges; adjusting statistical annotations in each figure based on recalculated metrics; and ensuring that all captions and figure references in the manuscript text

accurately describe the updated content.

We revised the Methods section to describe the improved filtering approach and updated workflow, updated the Results and Discussion sections to incorporate changes in interpretation based on revised validation outcomes, and adjusted conclusions to reflect updated findings.

Regarding the mentioned graph, the Y-axis labels were inconsistent with the caption: both panels displayed SD even though one was intended to show SWE. To avoid potential confusion and ensure clarity, we have removed the figure.

4. Point 3 from Ref #2, I am not sure the authors have addressed the point made, and I would like the referee's input there. There is a also new figure, but if these numbers are of averages for whole basins, that is not conveyed by the Figure text, as it is now. I would need the referee's comment on whether they are satisfied with the response here.

**Response:** We have clarified in the figure caption that the figure represents basin-averaged values in response to Point 3 from Referee #2. The updated caption now reads as follows:

[Figure]

Figure 7. Time series comparison of C-snow products with ASO observations averaged across multiple basins in California and Colorado at (a) 1-km, (b) 10-km, and (c) 25-km scales. The red points represent the average daily C-snow SD in all the selected

basins, and the different blue symbols indicate the daily average ASO data in various basins.

5. Point 4 from Ref #2 I see your thought process in the response, but I cannot see where you have changed the text, and so we will need to evaluate your response once you resubmit your manuscript.

**Response:** All the changes are marked in the revised manuscript in red. In response to Point 4 from Ref #2, we revised the manuscript as follows:

Section 2.3 (Auxiliary data): We added a detailed description of the ESA WorldCover 10 m 2020 land cover dataset.

Section 4 (Discussion): We included an analysis of the effect of the presence of permanent ice on the accuracy of the C-snow product. Permanent ice can cause significant overestimation in C-snow retrievals because of its snow-like electromagnetic properties and dynamic surface changes, necessitating quality control to filter such areas, especially at coarse scales. The observed performance in areas with moderate standard deviations of elevation (50–100 m) and moderately to highly forested areas (0.4–0.8) may not be solely attributable to one factor. The interaction between land cover types, topography, and environmental factors could lead to different retrieval accuracies.

Section 2.4 (Methodology): We added details on how elevation differences between stations and grid cells were calculated.

6. Point 5 I see that the authors include the revised text. I encourage the authors to re-read their text with fresh eyes to see if the language flows and the text clearly expresses a deeper analysis. For example, I think the first sentence "The differences in validation results of the C-snow product at different scales are due to the combined effects of various factors...." could be written in a more explicit way, such as naming those factors. Otherwise it is a rather vague sentence. Here also it would be good to have Ref #2's feedback on whether they are satisfied with the changes. There is also some grammar to correct in this new text as well.

**Response:** In the revised Discussion section, we have comprehensively reviewed and refined the text to improve clarity, analytical depth, and language flow. For example, we replaced vague statements with explicit descriptions of the underlying factors. The previous first sentence, "The differences in validation results of the C-snow product at different scales are due to the combined effects of various factors...", was rewritten to explicitly name these factors, i.e. validation dataset type (station vs. ASO), spatial representativeness, terrain complexity, land cover characteristics, and scale-related

aggregation effects.

Meanwhile, a thorough revision of the manuscript was performed by a native English speaker.

We thank Referee 1 for the constructive and thoughtful comments, which have helped us improve the manuscript. We have responded to the comments by presenting the original comments in black, our responses in blue, and the revisions in red.

**Referee #1**

This article conducts a thorough evaluation of the multi-scale performance of the Sentinel-1 SAR snow depth product, offering notable value in data validation and environmental impact analysis, with clear figures and fluent expression. However, its focus on validating existing algorithms rather than achieving a breakthrough limits its innovation, and certain language expressions lack smoothness. It is recommended that the manuscript be revised prior to submission for publication.

**#Major**

1. Although the article systematically evaluates the C-snow product across multiple scales, its core methodology—such as the C-band SAR-based snow depth retrieval algorithm—is not original to this study but builds upon prior work by Lievens et al. (2019). The innovation here lies primarily in data validation and scale analysis, yet these aspects do not represent a novel breakthrough in the field of remote sensing. It is recommended that the study explicitly highlight its unique contributions, such as whether it proposes a new scale-effect model or an improved retrieval method.

**Response:** We appreciate the reviewer's suggestions. The C-snow product used in this study is based on previous work by Lievens et al. (2019). However, our primary focus is on the multiscale validation and analysis of the C-snow product, with a particular emphasis on understanding the impact of scale effects on SD retrieval accuracy. While we do not introduce entirely new algorithms or models, we believe that the study offers valuable insights into the performance of the C-snow product across different scales. We revised the manuscript to highlight our contributions more explicitly.

(1)  To date, the C-snow dataset has been evaluated only from point to regional scales and not at the global scale (Lines 73−79). Our study comparatively assessed the C-snow dataset globally simultaneously by using station-based measurements and airborne LiDAR observations (Lines 94−96).

(2)  Multiscale C-snow datasets at 1, 10, and 25 km have been used to provide reference data to train machine learning models, improve passive microwave-based retrieval, and calibrate many hydrological models (Lines 79−81). However, existing validation articles focus only on the 1 km C-snow dataset and never assess the upscaled 10 and 25 km C-snow retrievals. We conducted a systematic assessment of C-snow products at three scales (1, 10 and 25 km).

(3)  We also provided a multiscale analysis of C-snow products and show that the

scale patterns vary across resolutions, which can enhance our understanding of the C-snow retrieval algorithm and validation work (Lines 89–92).

2. The introduction provides a detailed review of the development of SAR and microwave remote sensing in snow depth monitoring but fails to adequately justify the selection of the 1, 10, and 25 km scales for analysis or clarify their relevance to practical applications, such as hydrological modeling.

**Response:** Thank you for your constructive comments. The applications of SD data vary significantly depending on spatial resolution.

High-resolution SD data at 1 km are suitable for hydrological modeling and snow disaster monitoring (Wan et al., 2022). SD data at a 10 km resolution are appropriate for operational environmental prediction, hydrological forecasting and seasonal forecasting at regional scales (Alonso-Gonzá lez et al., 2018), whereas 25 km resolution data are widely used for SD monitoring, climate change analysis, and model evaluation at global and regional scales (Tanniru et al., 2023).

We mentioned that the C-snow product at 10 km and 25 km resolutions has already been used as a reference dataset, such as for training machine learning models to improve passive microwave SWE estimates (Lines 79–81). However, the accuracy of the 1 km C-snow product at these resolutions is still unknown. Moreover, the performance of the 1 km C-snow product at 10 km and 25 km resolutions is crucial for demonstrating whether active microwave remote sensing can provide a reliable reference SD dataset for passive microwave remote sensing.

**#minor**

1. Lines 20-21: Sentence is too long. "The results indicate that the scale patterns of the C-snow products across various resolutions differ from those of station- and airborne-based reference data." → "The scale patterns of C-snow products vary across resolutions. They differ from patterns observed in station and airborne reference data."

**Response:** The sentence has been revised as recommended.

**Revision:** The scale patterns of C-snow products vary across resolutions. They differ from the patterns observed in the station and airborne reference data.

2. Lines 366-367: The text contains redundant expressions, and optimization is recommended. "bias values ranging from -91.31 to -52.73 cm and ubRMSE values ranging from 104.3 to 83.29 cm"→"bias values decrease from -91.31 cm to -52.73 cm, while ubRMSE decreases from 104.3 cm to 83.29 cm".

**Response:** The text has been revised as recommended.

**Revision:** Compared with the airborne ASO data, the C-snow product became increasingly more accurate as the spatial scale increased, with bias values decreasing from -91.31 cm to -52.73 cm and ubRMSE decreasing from 104.3 cm to 83.29 cm.

3. Line 379: analized → analyzed.

**Response:** Corrected.

4. Suggestions for unified terminology. "ground-based measurements", "station observations"

**Response:** The manuscript has been revised to use "station observation" consistently.

We thank Referee 2 for the constructive and thoughtful comments, which helped us to improve the manuscript. We have provided our response to the comments, with the original comments in black text, our response in blue, and our revisions in red.

**Referee #2**

**General Comments:**

This paper addresses an important topic by evaluating the scale-dependent performance of the C-SNOW Sentinel-1 snow depth product against both in-situ measurements and airborne LiDAR observations. The multi-scale analysis and inclusion of geographic and land cover effects are valuable contributions to the remote sensing and snow hydrology communities. However, the manuscript would benefit from greater clarity in its introduction and methodological explanations, Additionally, some of the comparisons between datasets are not well-aligned in terms of spatial or temporal scale, which limits the interpretability of the findings. The Discussion section leans heavily on restating results rather than offering critical insight into the causes and implications of observed discrepancies. Overall, the paper has the potential to contribute knowledge on the accuracy of Sentinel-1 snow depth at different spatial scales, but revisions are needed to improve its structure, clarity, precision, scientific depth, and accuracy.

**Response:** We sincerely appreciate your valuable comments and suggestions on our manuscript. Your comments provided important guidance for us to improve the quality of the paper. In response to the issues you have raised, we have made the following main revisions:

• **We have revised the introduction to clarify the motivation and highlight the novelty of our study.**

We have rephrased the general statements to make them more specific and concise, added appropriate references to support key claims and provide a stronger theoretical foundation, and clearly articulated the research gap and explained how our study addresses it.

• **We have added a description of the land cover data in the Auxiliary data section.**

We have clarified the data source of the land cover data, elaborated on its classification, and explained how these data are utilized in this study.

• **We revised the description of the issue to clarify that some dataset comparisons are consistent at spatial and temporal scales.**

In the main text, we have explicitly clarified the comparison regions, which are consistent across both temporal and spatial dimensions.

• **We moved the figures and the corresponding results from the Discussion section to the Results section, and rewrote the Discussion section as suggested.**

The Discussion section was rewritten to provide a more in-depth analysis. It now includes a comparison with previous studies, an explanation of the differences in validation results at different scales, a discussion on the challenges of spatial scale conversion when station data are used for validation, the impact of permanent ice on the accuracy of the C-snow product, and directions for future research to address the limitations and improve the applicability of the C-snow product.

• **We have reorganized the figures in the revised manuscript.**

The main revisions include changing the color of the SNOTEL dataset in Figure 1 for better distinction; combining Figures 1 and 2 into a single figure with 4 panels (a–d); removing Figure 3; adding borders to markers, using basin codes instead of full names, clarifying the figure description for Figure 4; clarifying the purpose of the workflow in Figure 5; removing Figure 7; modifying Figure 8 to show the average time series across all stations with line plots; changing the caption of Figure 12 to clarify the representation of markers and bars; changing the histograms to pie charts in Figure 13; modifying the drawing of Figure 14 for more detail; retaining Figure 15, as it serves a different purpose than Figure 8 does; retaining Figure 16, as it provides insights into scale effects; and moving Figure 17 to the Results section.

**Major Comments:**

1. The rationale behind the use of Sentinel-1 data at 10 km and 25 km resolutions is unclear. If the primary aim of the paper is to evaluate the spatial and temporal performance of Sentinel-1 for snow monitoring, then standard higher-resolution products (e.g., 0.5 km and 1 km) would suffice. The inclusion of coarser resolutions needs to be better motivated and should be clearly stated and supported by appropriate literature and methodological context.

**Response:** Thank you for your constructive comments. The applications of SD data vary significantly depending on spatial resolution. High-resolution SD data at 1 km are suitable for hydrological modeling and snow disaster monitoring (Wan et al., 2022). SD data at a 10 km resolution are appropriate for operational environmental prediction, hydrological forecasting and seasonal forecasting at regional scales (Alonso-González et al., 2018), whereas 25 km resolution data are widely used for climate change analysis, and climate model evaluation at global and hemisphere scales (Tanniru et al.,

2023). The selection of these two lower resolutions is primarily based on two considerations. First, existing studies have already employed C-snow products at 10 km and 25 km resolutions as reference datasets, such as for training machine learning models to improve passive microwave SWE estimates (Xiong et al., 2022; Yang et al., 2024). These applications highlight the significance of evaluating the performance of C-snow products at these coarser resolutions to ensure their reliability and applicability in different contexts. Second, the performance of the 1 km C-snow product at 10 km and 25 km resolutions is crucial for demonstrating whether active microwave remote sensing can provide a reliable reference SD dataset for passive microwave remote sensing.

2. Since NRCS SNOTEL provides direct snow depth measurements, please clarify why a conversion from SWE to SD was performed. A fixed density to go from SWE to depth will impact the S1 evaluation, which is another consideration. Also, explicitly explain the choice of using a fixed snow density value of 0.24 g/cm³ for the Russia-SWE dataset. Is this based on regional averages or prior literature? Including a brief rationale would improve clarity.

**Response:** Thank you very much for your valuable comments. We have comprehensively updated all the results using the revised SNOTEL SD measurements. This update involved reprocessing the dataset with improved filtering criteria, rerunning the validation analyses, and integrating the updated SD values into the combined assessment with other datasets. Specifically, we revised Figures 1, 3, 4, 5, 8, 9, 10, 11, 12, 13, 14, 15, 16, and Figure A2, along with all the corresponding text descriptions, statistical analyses, and discussion sections related to these figures. We converted the Russia-SWE to SD using a snow density of 0.24 g/cm$^3$ on the basis of Takala et al. (2011) and Luojus et al. (2021).

3. In Figure 10, the comparison of the C-SNOW time series with spatially distributed ASO LiDAR data appears unrelated comparison. Since C-SNOW is also available as a spatial product over much of the Northern Hemisphere, it would be more appropriate to extract the C-SNOW spatial data closest to date to the ASO flight and perform a spatially explicit comparison. This would enable more meaningful evaluation and avoid misleading conclusions from mixed-scale comparison. Furthermore, the inclusion of air temperature in this figure is puzzling. The manuscript does not provide a rationale in the methodology for using temperature as a covariate or validation proxy. Since it is not used quantitatively in the analysis, I recommend removing it unless a clear scientific justification is provided. Overall, please revise this section to focus on meaningful spatial comparisons (e.g., using representative basins), and consider summarizing the evaluation using average statistics or appropriate spatial accuracy

plots (scatter, bias maps, etc.).

**Response:** In this figure, we extracted C-snow data within the basins in California and Colorado rather than for the entire Northern Hemisphere Mountain regions and analyzed the time series with ASO observations from different basins. To avoid confusion, we added a sentence to the caption stating that the red points represent the average daily C-snow SD across all the selected basins. The amount of data for individual basins was very small; thus, the comparisons in this section were based on all the selected basins. There was no quantitative use of temperature in the analysis, and we removed it.

**Revision:**

[Figure]

Figure 7. Time series comparison of C-snow products with ASO observations averaged across multiple basins in California and Colorado at (a) 1-km, (b) 10-km, and (c) 25-km scales. The red points represent the average daily C-snow SD in all the selected basins, and the different blue symbols indicate the daily average ASO data in various basins.

4. There are several issues in section 3.3. First, the phrase "other types" of land cover should be clarified—please specify which land cover categories are included beyond forest and permanent ice. Additionally, while the inclusion of permanent ice regions is noted, it's important to question the relevance of evaluating C-snow accuracy in such areas. Does the Lievens et al. algorithm or Sentinel-1 backscatter perform reliably in

permanent ice environments and dense forests? Please justify why this analysis was included and consider whether they should be treated as known limitations of the remote sensing platform and dataset. In continuation, some findings are counterintuitive and warrant further explanation. For instance, significant overestimation in areas with low forest fraction (0–0.2, figure 12c) is unexpected, as reduced vegetation cover typically enhances radar retrieval accuracy. Similarly, while elevation and elevation variability appear to influence performance, the paragraph lacks synthesis on why certain ranges (e.g., moderate forest and elevation variability) yield better results. Please also clarify how elevation differences between stations and grid cells were calculated, and how these mismatches propagate error across scales (this can go in methodology).

**Response:** We apologize for the confusion. We have added a description of the land cover data in the Auxiliary data section. The land cover data are from the ESA WorldCover 10 m 2020 product, which contains 11 land cover classifications, such as tree cover, shrubland, grassland, cropland, built-up, bare or sparse vegetation, snow and ice, permanent water bodies, herbaceous wetland, mangroves, moss and lichen. In this study, the tree cover type is labeled "tree cover", the snow and ice types are labeled "permanent ice", and all the remaining types are labeled "other type".

We analyzed the C-snow accuracy in permanent ice regions to provide a comprehensive assessment of the performance across different land cover types. This inclusion was driven by the need to understand the limitations and potential biases of the C-snow product in various environments, including those with permanent ice and dense forest cover. As mentioned in our manuscript, the C-snow product indeed shows several abnormally overestimated results in permanent ice regions, especially at larger scales (10- and 25-km resolutions). The retrieval performance in glaciated and dense forest cover areas needs investigation. Our current analysis serves as an initial evaluation of the performance in these challenging environments and highlights the need for future work to address these limitations.

Reduced tree cover can lead to lower attenuation of backscatter signals because of decreased vegetation density. However, the observed overestimation in these areas may not be solely attributable to tree cover. Other factors, such as surface roughness and the presence of other land cover types, could also play a significant role in influencing radar backscatter signals. The performance of the C-snow product appears to be influenced by elevation and its variability, as indicated by the varying Rbias values across different elevation ranges and standard deviations of elevation. While we have observed that the C-snow product performs best in areas with moderate standard deviations of elevation (50–100 m) and moderately to highly forested areas (0.4–0.8),

the underlying mechanisms are likely complex and multifaceted. The interaction between tree cover, topography, and environmental factors could lead to different retrieval accuracies.

In the Methodology section, we have clarified how elevation differences between stations and grid cells were calculated. Specifically, we computed the difference in elevation by subtracting the mean elevation of the grid cell (obtained from the high-resolution DEM) from the elevation of the station. This mismatch reflects the point-to-area scale representativeness issue, particularly in mountainous regions where snow properties vary significantly with elevation.

5. The current Discussion section is written more in the style of a results narrative. While the detailed reporting of accuracy metrics, grid-level behavior, and ice-related overestimates is important, most of the text focuses on what was found rather than interpreting what it means. To strengthen this section, I recommend separating the descriptive content into the results section and expanding the discussion with deeper analysis. For example, lay error analysis background, consider critically why ASO and station-based validation trends differ, what the implications of spatial representativeness are for coarse-scale validation, and how the known limitations of Sentinel-1 in forested or glaciated areas affect the broader applicability of C-snow. Comparisons with previous studies and a more explicit articulation of limitations and future directions would also help to better contextualize the results.

**Response:** We moved the figures and the corresponding results to the Results section and rewrote the Discussion section as suggested.

[revised manuscript text omitted]

**Minor Comments**

1. The introduction is currently in a general tone. Several statements are made without citing relevant studies or offering clear justification. Please consider grounding key claims with references and clarifying the motivation and novelty of the study more explicitly.

**Response:** We have revised the introduction to clarify the motivation and highlight the novelty of our study. Specifically, we have rephrased the general statements to make them more specific and concise, added appropriate references to support key claims and provide a stronger theoretical foundation, and clearly articulated the research gap and explained how our study addresses it.

2. Land cover classifications such as forest and permanent ice are introduced in the Results section without prior explanation. These should be defined and justified in the Methodology section, including the source of the land cover data and how the categories were used in the analysis.

**Response:** We have added a description of the land cover data in the Auxiliary data section. The land cover data are from the ESA WorldCover 10 m 2020 product, which contains 11 land cover classifications, such as tree cover, shrubland, grassland, cropland, built-up, bare or sparse vegetation, snow and ice, permanent water bodies, herbaceous wetland, mangroves, moss and lichen.

In this study, the tree cover type is labeled "tree cover", the snow and ice types are labeled "permanent ice", and all the remaining types are labeled "other type".

Therefore, we subsequently analyzed the effect of land cover on the accuracy of C-snow at different scales across these three labeled types.

3. Sections 3.3 and 3.4 both focus on geographic and environmental influences on snow depth retrieval. Merging them into a single cohesive section would improve readability and thematic consistency by presenting all location-based findings together.

**Response:** We have combined Sections 3.3 and 3.4 into a single section. The revised section presents all the findings related to geography and environmental factors in a more organized and clearer way.

4. Reorganize Figures and reconsider adding all figures for Improved Flow

a. Figure 14 could be merged with Figure 2 to streamline the presentation of the study area and grid setup. Label the nested grid structure directly in the map and refer to it when introducing the experimental design.

b. Figure 16 is conceptually related to Figure 12. Placing them closer together would enhance the narrative flow and allow readers to better understand the progression of results.

c. Please consider reducing the number of figures and balancing the proportion of figures and text. Refer to comments that suggest either removing or combining figures.

**Response:** We considered all the comments on the figures to enhance the overall flow. We referenced the comment to combine Figures 1 and 2, and as a result, we decided not to combine Figure 14 with Figure 2.

Conceptually, Figure 16 is related to Figure 12. However, Figure 16 focuses on the conditions of the three selected grids, while Figure 12 indicates the results across the mountainous region of the Northern Hemisphere. Thus, we did not place them together.

We considered the suggestions to balance the proportion of figures and text and removed some figures that contributed less useful information to this study.

**Line-to-Line Comments**

23 and elsewhere: The errors statistics are reported to the hundredth of a cm, but this level or precision is not realistic or warranted. Please consider the significant units here and elsewhere when reporting errors.

**Response:** Given that the station-observed SD, C-snow SD, and ASO SD values inherently have a precision that is finer than one-hundredth of a centimeter, we considered it appropriate to retain this precision in reporting error statistics.

26: Remove "Especially an".

**Response:** Done.

35: To clarify how snow depth (SD) information contributes to water availability, state "…estimated from snow depth (SD) and snow density.".

**Response:** Done.

**Revision:** The snow water equivalent (SWE) is a parameter that reflects how much water the snowpack contains and can typically be estimated from snow depth (SD) and snow density.

44: Add "one or several" before "meters".

**Response:** Added.

**Revision:** The snowpack in mountainous areas is typically deep (up to one or several meters), …

45: These density values are too high for most seasonal snow and in the range of when snow transitions to firn. Recommend revising to the typical range of density values (~100 to 550 kg/m3) for seasonal snow (see Sturm et al., 2010, J. Hydrometeorology).

**Response:** We revised the text to focus on the more typical range for seasonal snow (100–550 kg/m$^3$). We retained higher density values (550–700 kg/m$^3$) as a case to illustrate that under accumulation and compaction conditions, seasonal snow can reach such a density.

**Revision:** For example, the snow density typically ranges from 100 to 550 kg/m$^3$ for seasonal snow (Sturm et al., 2010). Owing to snowfall accumulation and prolonged wind- and gravity-driven compaction, it can reach 550–700 kg/m³ (Lemmetyinen et al., 2016; Venäläinen et al., 2021).

38- 50: The paragraph introduces the use of microwave remote sensing for SWE retrieval and suggests that SAR offers advantages over passive microwave techniques. However, it would benefit from greater specificity and clarity. For instance, clearly states that while passive microwave remote sensing is widely used, its coarse spatial resolution (~25 km) limits its ability to capture the fine-scale spatiotemporal variability of snowpacks in complex mountainous terrain. This will help establish a stronger context for the discussion of SAR advantages. Additionally, when discussing mountain snowpack complexities (lines 45–50), it would be helpful to explicitly connect these challenges—such as variable snow density, grain size, wind, and gravity-driven compaction, elevation, and aspect—to the limitations of passive microwave sensing. You may also consider briefly noting the difficulty of ground-based observations in remote, high-elevation regions, which further highlights the value of satellite-based

approaches. These additions will create a smoother transition to the next paragraph (lines 51–61), which focuses on SAR. Lastly, in line 43, replace "snow cover" with "snowpack" for technical accuracy.

**Response:** We have made several improvements to increase the clarity and effectiveness of our analysis. First, we added the difficulty of ground observations. Second, we clarify the limitations of passive microwave data, explicitly stating the coarse resolution constraint and linking it to the variability of mountain snowpack. Finally, we strengthened the advantages of synthetic aperture radar by emphasizing its superior resolution and suitability for complex terrain.

**Revision:** Conventional SD monitoring methods, such as manual field measurements and ground station observations, can provide accurate local data but are difficult to implement in remote mountainous areas with complex terrain. Microwave remote sensing is the most widely used technology for retrieving SWE because of its ability to penetrate snowpack and the volume scattering effects caused by snow particles (Chang et al., 1987; Tsang et al., 2022). While passive microwave remote sensing (e.g., radiometer-based methods) is typically employed, its coarse spatial resolution (~25 km) limits its ability to capture fine-scale spatiotemporal variability in snowpack properties, particularly in complex mountainous terrain.

51: Delete "the" before "monitoring".

**Response:** Deleted.

51-52: While this paragraph highlights recent advances using C-band SAR for SWE monitoring, it would be helpful to first acknowledge earlier foundational studies that have explored SAR for retrieving snow characteristics such as (Ulaby and Stiles, 1980; Bernier et al., 1999; Shi and Dozier, 2000; Chang et al., 2014; Lievens et al., 2019), there are many more. Including a broader set of references would better reflect the extensive work by the remote sensing community and provide context for the transition to C-band applications.

**Response:** We have revised the text to include a broader set of references, as suggested. Specifically, we have added Ulaby and Stiles (1980), Bernier et al. (1999), Shi and Dozier (2000), Chang et al. (2014), and Lievens et al. (2019).

**Revision:** In recent years, the scientific community has increasingly focused on monitoring the SWE in mountain regions using C-band SAR observations because of their strong penetration depth and data accessibility. Early studies on C-band SAR for SD estimation were limited primarily to shallow snow environments outside mountainous regions and co-polarization measurements, which showed limited

sensitivity to dry snow conditions (Bernier et al., 1999; Shi and Dozier, 2000). Theoretical advances in microwave scattering models (Ulaby et al., 1982; Chang et al., 2014) have improved the understanding of snowpack interactions with C-band signals.

52-54: The sentence discussing snow volume scattering being stronger in Ku-band compared to "other bands" could be made clearer. Please specify which bands are being referenced (e.g., X-, C, or L-band) and provide clearer citations. If this theoretical point is derived from Rott et al. (2010) or others, please state this explicitly and consider rephrasing for clarity and precision.

**Response:** We have clarified the term "other frequency bands" and revised the text accordingly.

**Revision:** Notably, although snow volume scattering is stronger in high-frequency Ku-bands than in other bands (e.g., X-, C, or L-band) in theory, the sensitivity of the backscattering coefficient at this frequency is also limited to approximately 150 cm (Rott et al., 2010; Cui et al., 2016; Zhu et al., 2021).

55-56: Instead of using "VH/VV," which may be unclear to some readers, consider using the more descriptive phrase "cross- to co-polarization ratio."

**Response:** The "cross-polarization ratio VH/VV" term has been used.

56-59: To avoid redundancy, consider replacing the second use of "notably" with an alternative phrase. Additionally, rephrase "due to the non-spherical properties of snowpack" to "due to the anisotropic nature of snow grains," which is a more accurate physical description. You may also cite relevant literature on this point—Lievens et al. (2022) references several useful studies that could support this claim.

**Response:** We changed the word "notably" to "in particular", rephrased the sentence, and added related references to the main text.

**Revision:** In particular, the backscattering coefficient at cross-polarization is more sensitive to volume scattering than co-polarization is because of the anisotropic nature of snow grains (Du et al., 2010; Chang et al., 2014; Leinss et al., 2016), and this physical mechanism is used for SD retrieval; in addition, co-polarized and cross-polarized signals are similar for surface scattering at the snow–soil boundary (Shi and Dozier, 2000; Lievens et al., 2022; Borah et al., 2024).

62-64: It would be helpful to clarify that the original C-snow product developed by Lievens et al. (2019) was produced at 1 km resolution without wet snow masking. In later studies, higher resolution (e.g., 500 m) products were developed with wet snow flagging. Mention explicitly that there are no C-Band products at 10 km and  25 km

resolution. Additionally, before introducing subsequent evaluation studies, please report the original study's metrics (e.g., RMSE, bias) from the original Lievens et al. study. This will help establish a clear baseline and better contextualize the results from later evaluations, thereby strengthening the motivation for your analysis.

**Response:** We have explicitly stated that the original C-snow product (Lievens et al., 2019) provides SD retrievals at a 1 km resolution without wet snow masking, and we have included the original performance metrics. No native C-band snow products exist at 10 km or 25 km resolution. The coarser-resolution datasets (10 km, 25 km) mentioned were derived from the original 1 km C-snow product through aggregation. We have now explicitly emphasized this distinction in the revised text.

**Revision:** The C-snow product provided SD retrievals at a 1-km resolution without wet snow masking (Lievens et al., 2019). It reported a temporal correlation ranging from 0.65 to 0.77 and a mean absolute error of 0.18–0.31 m. . . In addition, C-snow SD data at 10- and 25-km resolutions, which are derived from the 1-km C-snow product, have been used as reference datasets for training machine learning models to improve passive microwave SWE estimates (Xiong et al., 2022; Yang et al., 2024).

67-68: Please clarify that Hoppinen et al. (2024) evaluated the performance of the C-snow retrieval algorithm using airborne LiDAR data across two separate water years: 2020 and 2021. As currently written, "2020–2021" may be misinterpreted as a single water year.

**Response:** We revised the text to avoid misunderstanding.

**Revision:** Hoppinen et al. (2024) evaluated algorithm performance at six study sites across the western United States (US) using airborne LiDAR observations collected during the winters of 2019–2020 and 2020–2021, with mean RMSE and bias values of 92 cm and -49 cm, respectively.

69-71: The statement about Broxton et al. (2024) and Yang et al. (2024) using C-snow products at 10 and 25-km resolutions appears to be inaccurate. Broxton et al. used machine learning to enhance C-snow snow depth estimates at 0.5 and 1 km resolutions and compared these to University of Arizona SWE and airborne LiDAR data. They did not use C-snow to improve passive microwave SWE. Additionally, the studies you cited used either the 1 km or 500m C-snow product—not the 10 or 25 km. Please check for accuracy and revise.

**Response:** We removed the citation to Broxton et al. (2024).

**Revision:** In addition, C-snow SD data at 10- and 25-km resolutions, which are derived from the 1-km C-snow product, have been used as reference datasets for training

machine learning models to improve passive microwave SWE estimates (Xiong et al., 2022; Yang et al., 2024).

72-73: Please rephrase the sentence about Lievens et al. (2022) for clarity and precision. They used Sentinel-1 backscatter observations to retrieve snow depth across multiple resolutions in the European Alps and evaluated retrieval performance. This distinction is important to avoid confusion between backscatter versus the derived SD observation also in the monitoring and evaluating SD at different resolutions.

**Response:** We appreciate the comments and have revised the sentence to clarify that Lievens et al. (2022) used Sentinel-1 backscatter observations for retrieval.

**Revision:** Lievens et al. (2022) employed Sentinel-1 backscatter observations to retrieve SDs across multiple spatial resolutions in the European Alps and evaluated the retrieval performance.

84-85: ASO data are spatially extensive than stations, but do not have wider global coverage (i.e., they are only available in the western U.S.). Please revise phrasing for accuracy.

**Response:** Thank you. We have revised the text to clarify that while airborne LiDAR provides spatially extensive measurements compared with point-scale stations, its coverage is currently limited to specific regions (e.g., the western U.S.).

**Revision:** The latter provides spatially extensive SD mapping, which is more extensive than that provided by station data, and its coverage remains within the western US, whereas the station data are valuable for characterizing the SD distribution and assessing snow heterogeneity.

90-91: Recommend using a citation such as "The first 1 km SD product based on C-band SAR, covering all mountain ranges in the Northern Hemisphere, was developed by Lievens et al. (2019). The dataset is publicly available through the C-SNOW project (C-SNOW, 2024)." Instead of using a link in the text.

**Response:** We have removed this link and revised the text accordingly. In accordance with the instructions on the C-snow data website, we acknowledge the use of the data by citing Lievens et al. (2019).

**Revision:** The first 1-km SD product based on C-band SAR, covering all mountain ranges in the Northern Hemisphere, was developed by Lievens et al. (2019). The dataset is publicly available through the C-SNOW project.

104-105: The Zenodo link should be replaced with a proper DOI citation. Also, revise the sentence for typos and formatting—for instance, the link includes a fragment

("#YdYE...") that should be removed.

**Response:** We have revised this link.

**Revision:** The CanSWE dataset from Canada includes data from 273 stations in mountainous regions and can be accessed via https://zenodo.org/records/5217044 (Vionnet et al. 2021).

106-112: Remove links and include proper citations.

**Response:** We have removed the links and included appropriate citations in the text.

**Revision:** The GHCN dataset includes data from 4,133 stations in mountainous regions and provides SD values worldwide (Menne et al. 2012). The China-SD dataset from the China Meteorology Administration includes observations from 744 stations in mountainous regions. The SNOTEL dataset was acquired from 677 stations in mountainous regions in the US (Serreze et al. 1999). The Russia-SWE dataset from former Soviet Union regions contains observations from 52 stations in mountainous regions (Bulygina et al. 2011), and it can be downloaded from the All-Russia Research Institute of Hydrometeorological Information − World Data Center. Additionally, the Maine-SD dataset for the Maine region includes information from 92 stations in mountainous regions; it can be accessed via Maine Geological Survey Data.

127-133: ASO is not a LiDAR mission, ASO is a company that conducts LiDAR flight surveys using an airborne laser scanner (ALS). The dataset is available from 2013-2019 and 2022 to present. Please check the 2.2.2 section for accuracy. Refer to NSIDC, ASO website, and Painter et al 2016 paper. Also, clarify the reasoning behind using 3m instead of 50m.

**Response:** We apologize for the confusion. We revised the text as: The ASO data provide high-resolution, spatially comprehensive measurements of SD, SWE, and snow albedo in mountain basins by combining airborne lidar, imaging spectrometry, and physically-based snow modeling. According to the ASO website, the period of 2019–2021 also includes observations. We conducted comparative tests at both 3 m and 50 m resolution and found no significant differences after resampling. A 3 m resolution was selected to better capture fine-scale snow distribution patterns.

130-134: Please clarify that only California and Colorado ASO surveys were used.

**Response:** We have revised the text accordingly to explicitly state this.

**Revision:** To assess and compare the accuracy of the C-snow product at different scales, we obtained 59 ASO maps (within California and Colorado) at a 3-m resolution from September 2016 to May 2019.

141-142: Replace the word "corresponding datasets" with a direct reference to Table 1 for clarity. Also, in line with earlier comments, avoid including direct links (e.g., to Google Earth Engine) in the main text.

**Response:** We have revised the text to directly reference Table 1 instead of using "corresponding datasets" and have removed the direct link to Google Earth Engine as suggested.

**Revision:** To evaluate the influence of land cover type, forest fraction, and topography (elevation and its standard deviation) on the accuracy of C-snow SD, we collected auxiliary datasets (Table 1) from the Google Earth Engine and processed them at various scales (1, 10, and 25 km).

164: Clarify whether this is the Pearson correlation coefficient or other.

**Response:** We have clarified that the correlation coefficient refers to the Pearson correlation coefficient.

**Revision:** Four evaluation metrics were used to assess the C-snow products: the Pearson correlation coefficient (corr.coe), bias, unbiased root mean square error (ubRMSE), and relative bias (Rbias).

169-170: Add a line that explains what this section is about. It starts suddenly without making any coherence.

**Response:** We have added an introductory sentence to explain the purpose of Section 3.1.

**Revision:** We evaluated the C-snow SD retrievals through comparisons with station-based measurements across different spatial scales.

178-182: This text (and Figure 7) does not add much to the study, as it is all obvious and expected behavior for considering different spatial scales of a dataset. I recommend removing this.

**Response:** Removed.

189-192: The term "wet snow season" should be replaced with "ablation" or "melt season" to better reflect the physical processes and limitations of Sentinel-1 during this period. The phrase "mismatch of snow season length" is vague. It's not clear whether the authors refer to differences in snow onset and melt timing, the overall duration of snow cover, or specific discrepancies, C-Band data is only available till the end of April anyway as Sentinle-1 is not reliable in ablation season.

**Response:** We have replaced "wet snow season" with "melt season" to better align

the physical processes and Sentinel-1 limitations during this period. We also acknowledge the ambiguity in the original phrasing and have clarified the seasonal comparison to focus on the contrast between dry snow accumulation and melt phases.

**Revision:** As the spatial scale increases from 1 to 25 km, both the magnitude and duration of the discrepancies between C-snow and station SDs increase. Specifically, the average SD from the stations becomes increasingly greater than the C-snow SD during the dry snow season and increasingly lower during the melt season.

195-196: This line should be part of the limitation section

**Response:** The sentence is closely related to the results discussed in the preceding text and serves as a transition to the subsequent text, where we emphasize the need to assess relevant spatially distributed influential factors.

223-225: What is the other type of land cover you are talking about here? Please explain how you expect Sentinel-1 to work in permanent ice regions, what are those regions?

**Response:** We have added a description of the land cover data in the Auxiliary data section, where the "other types" refer to regions without permanent ice cover and tree cover. In permanent ice regions, such as Greenland, the Hindu–Kush Himalayas, and the Rocky Mountains, there are also Sentinel-1 observations, and these observations can be used to retrieve the SD.

271-276: Here and elsewhere – what is the purpose of comparing a 10-km or 25-km estimate of snow depth with a station? One would not expect the station to match those very different scales.

**Response:** The C-snow product at 10 km and 25 km resolutions has already been used as a reference dataset, such as for training machine learning models to improve passive microwave SWE estimates. However, the accuracy of the 1 km C-snow product at these resolutions is still unknown. Moreover, the performance of the 1 km C-snow product at 10 km and 25 km resolutions is crucial for demonstrating whether active microwave remote sensing can provide a reliable reference SD dataset for passive microwave remote sensing.

We fully acknowledge that comparing coarse-resolution estimates with point-scale station measurements has inherent limitations because of scale discrepancies. This is precisely why we introduced comparisons with ASO data in our analysis. The ASO observations provide spatially extensive, high-resolution SD measurements that offer a more robust evaluation of the C-snow product.

282-283: What evidence do you have for this statement? I do not think that spatial representativeness is guaranteed for a station in a flat area. Additionally, I would not conflate low elevation with flat, as there can be topographic complexity/variability even at lower elevations.

**Response:** We agree that spatial representativeness cannot be guaranteed even in low-elevation areas, and we have revised our statement to be more precise. Our key point is that compared with high-elevation regions, the lower-elevation areas (<1000 m) in our study domain generally exhibit reduced topographic complexity (average slope <5°), lower spatial variability in snow distribution patterns, and more homogeneous land cover characteristics. These factors collectively contribute to better station representativeness at coarse scales in these areas than in high-elevation regions. We revised the text to emphasize the characteristics of low-elevation areas rather than simply describing them as flat.

**Revision:** For the first nested grid, the terrain is predominantly low-elevation (below 1000 m) at both the 10- and 25-km scales, which shows lower spatial variability than high-elevation regions do.

306-307: It should be part of the data section

**Response:** Moved.

**Figures and Table Comments**

Figure 1: Use distinct colors for SNOTEL and GHCN since some of the stations are close to each other and it is hard to distinguish.

**Response:** The color of the SNOTEL dataset has been changed for improve clarity.

Revision:

[Figure]

Figures 1 and 2: Consider combining these into a single figure with 4 panels (a-d).

**Response:** This figure has been combined with Figure 1 into a single figure with 4 panels (a–d).

Revision:

[Figure]

Figure 1. Spatial distribution of (a) stations in various SD datasets and of the matched grids at the (b) 1-km, (c) 10-km, and (d) 25-km scales. Zoomed-in views show the detailed distributions of grid locations in the Sierra Nevada range over the US and the Jotunheimen mountain range in Norway and Sweden.

Figure 2: Continent labels are hiding in some places. Make it consistent—no labels or labels everywhere.

**Response:** This is a default basemap, and continent labels cannot be removed individually. Although some labels are overlaid, this does not affect access to key information.

Figure 3: This figure may not be necessary, especially given the already high number of figures (19). However, if you choose to include it, consider showing ASO data coverage across the entire US and highlighting (e.g., with a box) the region used in your analysis. This would provide helpful context without redundancy.

**Response:** This figure has been removed.

Figure 4: Consider using station codes instead of full names to reduce clutter. Add borders to markers for better visibility and clarify the meaning of each color in the figure description.

**Response:** To improve visibility and clarity, we added borders to the markers, used basin codes rather than full names to minimize clutter, and added descriptions to the figure.

**Revision:**

[Figure]

Figure 2. Temporal distribution of the ASO observations used in this study. The markers with different colors represent different basins.

Figure 5: Please distinguish two boxes by panels for example say panel a and panel and state it in the figure description. Also, the arrow between accuracy analysis and uncertainty analysis is misleading. Are you trying to say uncertainty analysis comes after accuracy analysis? Regardless arrow is not needed.

**Response:** Thank you for your comments. We would like to clarify that the figure is intended to present a complete workflow rather than distinct panels; therefore, labeling the two boxes as "Panel a" and "Panel b" may not be appropriate. The arrow between "accuracy analysis" and "uncertainty analysis" illustrates the sequential flow of the analytical process.

Figure 7: It appears that Figure 7 is only showing C-snow data at different spatial scales (1, 10, and 25 km) across three mountain regions, without any comparison to reference or evaluation datasets. If that's the case, the purpose of this figure needs to be clearly stated—whether it is to demonstrate spatial variability or resolution effects. As currently placed, the figure does not align well with the section, which focuses on comparing C-snow with observed SD. Consider either relocating the figure to a more appropriate section or removing it if it doesn't directly contribute to the evaluation narrative.

**Response:** This figure has been removed.

Figure 8: Please clarify whether Figure 8 shows the average time series across all stations or if it represents individual station values. If it's an average, this should be explicitly stated in the caption and main text. However, averaging across diverse stations may mask site-specific dynamics and variability. Consider instead selecting one or a few representative stations to illustrate the temporal mismatch at different scales more clearly. Additionally, using a line plot instead of scatter points would improve readability and better highlight trends over time.

**Response:** Figure 8 shows the average time series across all the stations, and we added descriptions in the caption and main text. To present temporal trends more clearly, we used line plots to increase readability. A more detailed case study at the nested grid scale is shown in Figure 15, which shows how regional variability and site-specific conditions affect the consistency between C-snow and station observations at multiple scales.

**Revision:**

[Figure]

Figure 5. Average weekly SD time series of stations and corresponding C-snow grids at (a) 1-, (b) 10-, and (c) 25-km resolution across the mountainous regions of the Northern Hemisphere.

Figure 10: Check major comment section

**Response:** We modified the figure on the basis of the major comment.

Figure 12: It's not immediately obvious that the markers show actual SD values, and the bars show relative error. Adding a brief formula or explanation for Rbias in the caption or methods would also help.

**Response:** We have revised the caption of this figure to clarify that the bars represent the Rbias, while the markers show the average SD observed at stations and in the C-snow product.

**Revision:** Impact of different (a) station-observed SDs, (b) land cover types, (c) forest fractions, (d) elevations, (e) standard deviations of elevation, and (f) elevation differences between stations and grids on the accuracy of C-snow SD products across various scales. The bars indicate the Rbias between the C-snow and station-observed snow SDs, while the markers show the average SDs from the stations and the C-snow product. The left axis corresponds to Rbias, and the right axis corresponds to the

average SD.

Figure 13: The left plots may be better conveyed as pie charts rather than bar charts because they add up to 100%.

**Response:** The histograms were modified to pie charts.

**Revision:**

[Figure]

Figure 13. Pie charts (left column) and spatial distributions (right column) of Rbias at (a) 1-, (b) 10-, and (c) 25-km scales.

Figure 14: Refer to minor comments. Additionally, the layout of this figure is not ideal, as the three selected regions overlap the global map but instead could be shown separately with more detail.

**Response:** We referenced the comment to combine Figures 1 and 2 and therefore did not combine Figure 14 with Figure 2. Instead, we modified the drawing of Figure 14 to provide more detail.

**Revision:**

[Figure]

Figure 15: How figure 15 is different than Figure 8? Figure 15 makes more sense in section 3.1 in replacement of Figure 8.

**Response:** Figure 8 and Figure 15 serve different purposes in our analysis. A general comparison of the time series between the C-snow products and station measurements across the entire Northern Hemisphere mountainous region is shown in Figure 8, highlighting how the temporal correlation and seasonal mismatch vary with spatial scale. On the other hand, Figure 15 provides a more detailed case study at the nested grid level, examining how regional variability and specific station conditions affect the agreement between C-snow and station measurements at multiple scales. We have retained Figure 8 in the manuscript because it provides a necessary overview of the scale-dependent temporal trends across all the stations.

Figure 16: This figure does not contribute much to the study and can be considered for removal.

**Response:** We believe that this figure provides important insights into the scale effects associated with geographic heterogeneity. The differences in land cover, forest fraction, elevation, and elevation variability between the 10- and 25-km grids across the three nested regions are shown. Therefore, we have retained this figure in the revised manuscript.

Figure 17: It should be a part of the results.

**Response:** Moved.